# Logistic-Normal Likelihoods for Heteroscedastic Label Noise

**Erik Englesson**                                                    *engless@kth.se*
*KTH Royal Institute of Technology*

**Amir Mehrpanah**                                                    *amirme@kth.se*
*KTH Royal Institute of Technology*

**Hossein Azizpour**                                                  *azizpour@kth.se*
*KTH Royal Institute of Technology*

**Reviewed on OpenReview:** *https://openreview.net/forum?id=7wA65zL3B3*

## Abstract

A natural way of estimating heteroscedastic label noise in regression is to model the observed (potentially noisy) target as a sample from a normal distribution, whose parameters can be learned by minimizing the negative log-likelihood. This formulation has desirable loss attenuation properties, as it reduces the contribution of high-error examples. Intuitively, this behavior can improve robustness against label noise by reducing overfitting. We propose an extension of this simple and probabilistic approach to classification that has the same desirable loss attenuation properties. Furthermore, we discuss and address some practical challenges of this extension. We evaluate the effectiveness of the method by measuring its robustness against label noise in classification. We perform enlightening experiments exploring the inner workings of the method, including sensitivity to hyperparameters, ablation studies, and other insightful analyses.

## 1 Introduction

Supervised learning relies on datasets with input-label pairs, in which some labels are likely to be wrong, *e.g.*, due to annotation mistakes in a classification problem or measurement devices' precision in a regression problem. Even the systematically annotated datasets, like ImageNet, contain noisy labels (Beyer et al., 2020). It is, therefore, crucial to be able to effectively handle label noise.

**Heteroscedastic Noise in Regression: A Motivation.** A natural way to deal with mislabeled examples in regression is to model the observed target ($y$) as the true target ($\mu$) with additive noise ($\epsilon$) (Nix & Weigend, 1994; Kendall & Gal, 2017; Lakshminarayanan et al., 2017):

$$y(\boldsymbol{x}) = \mu(\boldsymbol{x}) + \epsilon(\boldsymbol{x}). \tag{1}$$

Assuming a normally-distributed $\epsilon$ with zero mean and variance $\sigma^2$, the *likelihood* of the observed target becomes $p(y|\boldsymbol{x}, \mu, \sigma^2) = \mathcal{N}(y; \mu(\boldsymbol{x}), \sigma^2(\boldsymbol{x}))$. Then, neural networks can be used to estimate the input-dependent parameters of the distribution: $\mu(\boldsymbol{x}) \approx \mu_{\boldsymbol{\theta}}(\boldsymbol{x})$, $\sigma^2(\boldsymbol{x}) \approx \sigma_{\boldsymbol{\theta}}^2(\boldsymbol{x})$. The parameters of the neural network, $\boldsymbol{\theta}$, are typically learned via some type of maximum likelihood estimation with N data samples:

$$\arg\max_{\boldsymbol{\theta}} - \sum_{i=1}^{N} \frac{(y_i - \mu_{\boldsymbol{\theta}}(\boldsymbol{x}_i))^2}{2\sigma_{\boldsymbol{\theta}}^2(\boldsymbol{x}_i)} + \frac{1}{2}\log\sigma_{\boldsymbol{\theta}}^2(\boldsymbol{x}_i) + const \tag{2}$$

This loss is of an interesting form: a label-dependent loss (mean square error) divided by the noise variance $\sigma^2$ and a regularizing term for $\sigma^2$ (log-partition). Hence, $\sigma^2$ acts as an inverse importance weight of the mean

squared error loss. For example, for high-residual examples, the penalty of the mean squared error loss can be reduced by increasing $\sigma^2$, leading to higher freedom for $\mu_{\boldsymbol{\theta}}$ to deviate from $y$.

We aim to obtain an analogously simple loss function to attenuate erroneous labels for *classification* tasks.

**Heteroscedastic Noise in Classification with Attenuation**. Kendall & Gal (2017) argued such a loss attenuation property is desirable for classification, and proposed to learn the mean and covariance of a normal distribution over the pre-softmax logits by maximizing a categorical likelihood. This results in a form of loss attenuation, but interestingly not the same as in regression.

**Contributions.**   The main contributions of our work are[1]:

- We propose a natural extension of the above regression noise model to classification and show that it leads to the target following a Logistic-Normal distribution (Atchison & Shen, 1980); see Section 2.2.

- We propose using the Logistic-Normal likelihood in a maximum a posteriori estimation setting and show its negative log-likelihood has the same desirable loss attenuation properties as in the regression case, *e.g.*, reducing the contribution of high-residual examples; see Sections 2.3 & 2.4. Furthermore, we address implementation challenges and propose practically important techniques; see Section 3.

- We empirically study the proposed loss on several datasets with synthetic and natural noise, where we show improved robustness to label noise compared to recent works; see Section 5.

## 2   Method

Our goal is to extend the simple and probabilistic loss attenuation approach from regression to classification. Here, we give a high-level overview of the problem and describe the main idea of our method. We give important details of our approach in Section 3.

### 2.1   Background and Problem

In this section, we define label noise, identify its source, and highlight the challenges it poses for deep learning.

**Dataset Generation.** In classification, we consider a dataset as samples from an unknown joint distribution: $\mathcal{D} = \{\boldsymbol{x}_i, y_i\}_{i=1}^N$, $(\boldsymbol{x}_i, y_i) \sim p(\boldsymbol{x}, y) = p(y|\boldsymbol{x})p(\boldsymbol{x})$. The generation process can be interpreted as, first, sampling the input, and then the label: (i) $\boldsymbol{x}_i \sim p(\boldsymbol{x})$, (ii) $y_i \sim p(y|\boldsymbol{x}_i)$. This is, in fact, how many multiclass datasets are constructed, *i.e.*, first collecting a large set of inputs and then, automatically or manually, annotating each input with a single output label.

**Noisy Labels.** The label $y_i$ can be seen as a categorical distribution via one-hot encoding, $\boldsymbol{\delta}_c$, which is equal to one in component $c$, and zero otherwise. As $y_i$ is a single sample from $p(y|\boldsymbol{x}_i)$, we see $\tilde{p}(y|\boldsymbol{x}_i) = \boldsymbol{\delta}_{y_i}$ as a crude estimate of the true label distribution $p(y|\boldsymbol{x}_i)$. Occasionally, sampling $p(y|\boldsymbol{x})$ gives an unlikely sample $y$, *e.g.*, when probable errors are made in the annotation, causing a large difference between $p(y|\boldsymbol{x})$ and $\tilde{p}(y|\boldsymbol{x})$. We aim to model this label noise (difference between $p(y|\boldsymbol{x})$ and $\tilde{p}(y|\boldsymbol{x})$) in the pre-softmax logit space.

**Learning with Label Noise.** Specifically, we are interested in a probabilistic model with parameterized output distribution $p_{\boldsymbol{\theta}}(y|\boldsymbol{x})$ and aim to optimize $\boldsymbol{\theta}$ so that the prediction is close to the true distribution, $p_{\boldsymbol{\theta}}(y|\boldsymbol{x}_i) \approx p(y|\boldsymbol{x}_i)$, using only the (noisy) training data $(\boldsymbol{x}_i, \tilde{p}(y|\boldsymbol{x}_i))$. Although we see both $p_{\boldsymbol{\theta}}(y|\boldsymbol{x})$ and $\tilde{p}(y|\boldsymbol{x})$) as approximations of $p(y|\boldsymbol{x})$, note that they differ in that the former is predicted, and the latter is directly determined by the given label. The challenge of learning from noisy labels with deep networks is their susceptibility to overfitting to $\tilde{p}(y|\boldsymbol{x}_i) = \boldsymbol{\delta}_{y_i}$, for which we propose the following method and noise model.

### 2.2   Modelling Label Noise with Logistic-Normal Likelihoods

The core idea of our work is: Given an invertible mapping from logit space to the probability simplex, we can define a regression noise model in logit space and use the map to get a noise model for classification. Here, we go over the map (softmax centered), the noise model, and the corresponding classification likelihood.

---

[1]Our code is available at: `https://github.com/ErikEnglesson/Logistic-Normal`

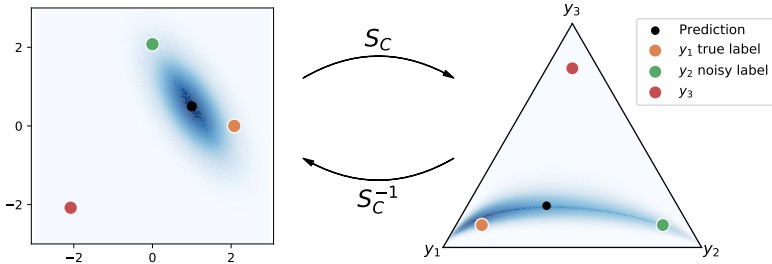

Figure 1: **Logit Space and Probability Simplex Equivalence.** An example showing how a normal distribution in $\mathbb{R}^2$ corresponds to a Logistic-Normal distribution in $\mathring{\Delta}^2$, when using the softmax centered function as the bijective transformation.

**Softmax Centered.** The softmax function $S(\boldsymbol{z}) = S([z_1, \ldots, z_K]) = [e^{z_1}, \ldots, e^{z_K}]/\sum_{i=1}^{K} e^{z_i}$ is a map from logit space $\mathbb{R}^K$ to the probability simplex of $K$ classes $\Delta^{K-1}$. This map is not invertible, as $S(\boldsymbol{z}) = S(\boldsymbol{z} + \boldsymbol{c})$ for any $\boldsymbol{c}$ in $\mathbb{R}^K$ where all components of $\boldsymbol{c}$ are equal. An alternative is the bijective softmax centered function $S_C(\boldsymbol{z}) = S([z_1, \ldots, z_{K-1}, 0])$ from $\mathbb{R}^{K-1}$ to the interior of the simplex, $\mathring{\Delta}^{K-1} = \{\boldsymbol{p} \in \Delta^{K-1} \mid p_i > 0, \forall i \in \{1, 2, \ldots, K\}\}$, with inverse $S_C^{-1}(\boldsymbol{q}) = \log([\frac{q_1}{q_K}, \frac{q_2}{q_K}, \ldots, \frac{q_{K-1}}{q_K}])$.

**Noise Model.** Consider a multivariate version of Equation 1 where $\boldsymbol{y}(\boldsymbol{x}), \boldsymbol{\mu}(\boldsymbol{x}), \boldsymbol{\epsilon}(\boldsymbol{x}) \in \mathbb{R}^{K-1}$ are vectors in logit space. We can apply the softmax-centered transformation to this noise model to get a label noise model for classification:

$$S_C(\boldsymbol{y}(\boldsymbol{x})) = S_C(\boldsymbol{\mu}(\boldsymbol{x}) + \boldsymbol{\epsilon}(\boldsymbol{x})). \tag{3}$$

With a deterministic $\boldsymbol{\epsilon}$, analogous to regression, we model the noisy label distribution $\tilde{p}(y|\boldsymbol{x}) = S_C(\boldsymbol{y}(\boldsymbol{x}))$ as the true label distribution $p(y|\boldsymbol{x}) = S_C(\boldsymbol{\mu}(\boldsymbol{x}))$ with additive noise $\boldsymbol{\epsilon}(\boldsymbol{x})$ in logit space[2]. Hereafter, we occasionally omit the dependence of variables $\boldsymbol{y}, \boldsymbol{\mu}, \boldsymbol{\Sigma}$ to $\boldsymbol{x}$ for notational convenience.

**Likelihood Function.** Assuming a normally distributed $\boldsymbol{\epsilon}(\boldsymbol{x})$ in Equation 3 with zero mean and covariance $\boldsymbol{\Sigma}(\boldsymbol{x})$, the *likelihood* of the observed target $\tilde{p}(y|\boldsymbol{x})$ is:

$$\frac{1}{\prod_{k=1}^{K} \tilde{p}_k(y|\boldsymbol{x})} \frac{1}{|(2\pi)^{K-1}\boldsymbol{\Sigma}|^{\frac{1}{2}}} e^{-\frac{1}{2}(S_C^{-1}(\tilde{p}(y|\boldsymbol{x}))-\boldsymbol{\mu})^T \boldsymbol{\Sigma}^{-1}(S_C^{-1}(\tilde{p}(y|\boldsymbol{x}))-\boldsymbol{\mu})}. \tag{4}$$

As now $\boldsymbol{\mu}(\boldsymbol{x}) + \boldsymbol{\epsilon}(\boldsymbol{x})$ is a Gaussian random variable, transforming it with the softmax bijection leads to a transformed random variable with a density proportional to $\mathcal{N}(S_C^{-1}(\boldsymbol{q}); \boldsymbol{\mu}, \boldsymbol{\Sigma})$ for $\boldsymbol{q} \in \mathring{\Delta}^{K-1}$ (derivation in Appendix B.1). Importantly, this corresponds to a well-studied probability density function, called *Logistic-Normal distribution* (Atchison & Shen, 1980), which is defined for categorical distributions in $\mathring{\Delta}^{K-1}$.

As the map is bijective, this model gives rise to dual interpretations: (i) a regression problem with Gaussian likelihoods with targets, $S_C^{-1}(\tilde{p}(y|\boldsymbol{x}))$, in logit space, or (ii) a classification problem with Logistic-Normal likelihoods with targets, $\tilde{p}(y|\boldsymbol{x})$, in the probability simplex. This duality is visualized in Figure 1.

### 2.3 Estimation with Logistic-Normal Likelihoods

We use deep networks with parameters $\boldsymbol{\theta}$ to predict true logit vectors $\boldsymbol{\mu}_i \approx \boldsymbol{\mu_\theta}(\boldsymbol{x}_i)$ and per-example (heteroscedastic) noise covariance matrices $\boldsymbol{\Sigma}_i \approx \boldsymbol{\Sigma_\theta}(\boldsymbol{x}_i)$. We use separate linear layers for $\boldsymbol{\mu}$ and $\boldsymbol{\Sigma}$ that share the same backbone. The network parameters are found by minimizing the following negative log-likelihood of

---

[2]Regarding $\tilde{p}(y|\boldsymbol{x}) = S_C(\boldsymbol{y}(\boldsymbol{x}))$, note that $\tilde{p}(y|\boldsymbol{x}) = \boldsymbol{\delta}_y \in \Delta^{K-1}$ corresponds to a corner of the probability simplex, and is therefore not in the co-domain of the softmax centered function. We solve this issue by slightly diffusing it with label smoothing: $\tilde{p}(y|\boldsymbol{x}_i) \triangleq (1 - t)\boldsymbol{\delta}_{y_i} + t\boldsymbol{u} \in \mathring{\Delta}^{K-1}$, where $t > 0$, and $\boldsymbol{\delta}_y$ and $\boldsymbol{u}$ are the delta and uniform distributions over $K$ classes, respectively. We use $t = 0.01$ in all our experiments, except for a sensitivity analysis in Appendix G.1.

the dataset (inputs $\boldsymbol{x}_i$ and observed targets $S_C^{-1}(\tilde{p}(y|\boldsymbol{x}_i)) = \boldsymbol{y}_i$) in addition to the negative log-prior over $\boldsymbol{\theta}$:

$$\frac{1}{2}\sum_{i=1}^{N}(\boldsymbol{y}_i - \boldsymbol{\mu}_i)^T\boldsymbol{\Sigma}_i^{-1}(\boldsymbol{y}_i - \boldsymbol{\mu}_i) + \log|\boldsymbol{\Sigma}_i| + const. \tag{5}$$

As the first factor in Equation 4 is independent of $\boldsymbol{\theta}$, it is part of the constant term. Our goal was to find a classification loss with similar loss attenuation to the loss for regression. We see that our loss in Equation 5 is almost identical to a multivariate version of the regression loss in Equation 2. Hence, the loss has the same attenuation effect: It can be decreased by learning $\boldsymbol{\Sigma}_i$ such that $(\boldsymbol{y}_i - \boldsymbol{\mu}_i)^T \boldsymbol{\Sigma}_i^{-1} (\boldsymbol{y}_i - \boldsymbol{\mu}_i)$ is smaller than $(\boldsymbol{y}_i - \boldsymbol{\mu}_i)^T(\boldsymbol{y}_i - \boldsymbol{\mu}_i)$, leading to higher freedom for $\boldsymbol{\mu}_i$ to deviate from $\boldsymbol{y}_i$ as the penalty is reduced.

We now established the *capability* for Logistic-Normal likelihood to attenuate high-residual samples and potentially predict their correct mean. Next, we do a gradient analysis whereby we show how such behavior is in fact *encouraged* when learning with gradient-following methods. In Section 5, we empirically verify the *realization* of such effect not only through the final performance on noisy training data but also with targeted analyses of the training behavior.

### 2.4 Loss Attenuation: A Gradient Perspective

Let $\mathcal{L}$ be the negative log-likelihoods in Equation 5, then the gradients w.r.t. $\boldsymbol{\mu}$ of example $j$ are:

$$\frac{\partial\mathcal{L}}{\partial\boldsymbol{\mu}_j} = -\boldsymbol{\Sigma}_j^{-1}(\boldsymbol{y}_j - \boldsymbol{\mu}_j). \tag{6}$$

This reveals an interesting form where the gradients are related to the difference between the target logit ($\boldsymbol{y}_j$) and the predicted mean ($\boldsymbol{\mu}_j$), and this difference is scaled by the inverse covariance matrix ($\boldsymbol{\Sigma}_j^{-1}$). This scaling is a major difference compared to the gradients of the negative log-likelihood of a categorical distribution (CE) w.r.t. its logits: $-(\boldsymbol{\delta}_{y_j} - S(\boldsymbol{z}_j))$, where $\boldsymbol{z}_j$ is the logits and $S(\cdot)$ is the standard softmax function.

To better understand what per-example $\boldsymbol{\Sigma}$ matrices the network tries to predict, we look at the optimal covariance matrix, *i.e.*, when $\frac{\partial\mathcal{L}}{\partial\boldsymbol{\Sigma}_j} = 0$ (details in Appendix B.2):

$$\boldsymbol{\Sigma}_j^{opt} = (\boldsymbol{y}_j - \boldsymbol{\mu}_j)(\boldsymbol{y}_j - \boldsymbol{\mu}_j)^T. \tag{7}$$

The geometric interpretation is that the optimal density is a thin hyperellipsoid with its center on $\boldsymbol{\mu}$ and highest variance in the direction of the noisy target $\boldsymbol{y}$, see Figure 1 left.

Plugging the optimal covariance matrix from Equation 7 in Equation 6, we get (details in Appendix B.3):

$$\left.\frac{\partial\mathcal{L}}{\partial\boldsymbol{\mu}_j}\right|_{\boldsymbol{\Sigma}_j^{opt}} = -\frac{(\boldsymbol{y}_j - \boldsymbol{\mu}_j)}{||(\boldsymbol{y}_j - \boldsymbol{\mu}_j)||_2^2}. \tag{8}$$

That is, for a given $\boldsymbol{\mu}_j$, the corresponding optimal covariance matrix divides the difference between the label and the logits by its squared l2-norm, which is exactly the loss attenuation property of our method. Clearly, the role of $\boldsymbol{\Sigma}(\boldsymbol{x})$ is to increase and decrease the gradients for low- and high-residual examples, respectively.

As gradients are not affected by constants, and as the LN and multivariate normal likelihoods are the same up to constants, the gradients here also apply to the regression case. This reflects the analogy of LN for classification, with the loss-attenuating properties of the regression case, that we were after in this work.

## 3 Important Details

While the essence of our proposed noise model and the induced likelihood function are simple at the high level, it involves important details that we discuss in this section.

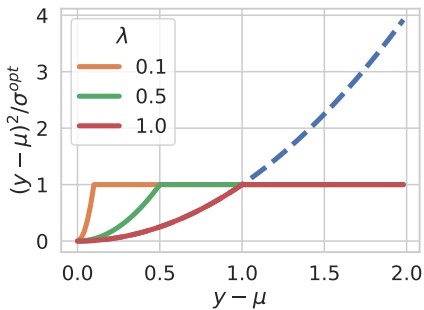 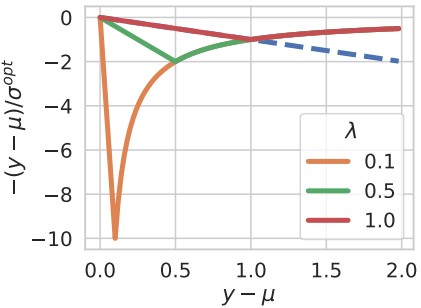

Figure 2: **Controlling Loss Attenuation with $\lambda$.** In higher dimensions, $\lambda$ is necessary to ensure invertibility of $\mathbf{\Sigma}$, but it is also a way to control the loss attenuation for low residual examples. We plot the loss (left) and gradients w.r.t. $\mu$ (right) with the optimal obtainable $\sigma$ (binary classification) against varying residuals. Clearly, the value of $\lambda$ determines the residual threshold for where loss attenuation starts. We have loss attenuation when the residual is larger than $\lambda$, resulting in a loss of one and a gradient of $-1/(y - \mu)$, and otherwise the loss is the mean-squared error loss divided by $\lambda^2$.

### 3.1 Estimating Per-Example Covariance Matrices

Note that the output covariance matrix $\mathbf{\Sigma}(\boldsymbol{x})$ has $\mathcal{O}(K^2)$ parameters, needs to be symmetric and semi-positive definite. Next, to feasibly predict $\mathbf{\Sigma}(\boldsymbol{x})$ for a large number of classes, we make use of the analysis for the optimal $\mathbf{\Sigma}(\boldsymbol{x})$ to propose a structured reparametrization requiring only $\mathcal{O}(K)$ parameters.

**Parametrization of $\mathbf{\Sigma}$.** Hinging on the rank deficiency of the optimal covariance matrix, we parametrize it as $\mathbf{\Sigma}_{\boldsymbol{\theta}}(\boldsymbol{x}) = \mathbf{\Sigma}_{\boldsymbol{\theta}}^{\frac{1}{2}}(\boldsymbol{x})\mathbf{\Sigma}_{\boldsymbol{\theta}}^{\frac{1}{2}}(\boldsymbol{x})^T$ where:

$$\mathbf{\Sigma}_{\boldsymbol{\theta}}^{\frac{1}{2}}(\boldsymbol{x}) = \boldsymbol{c}_{\boldsymbol{\theta}}(\boldsymbol{x})\boldsymbol{c}_{\boldsymbol{\theta}}(\boldsymbol{x})^T + \lambda\boldsymbol{I}, \tag{9}$$

with $\boldsymbol{c}_{\boldsymbol{\theta}}(\boldsymbol{x}) \in \mathbb{R}^{K-1}$, and a hyperparameter $\lambda \in \mathbb{R}_{>0}$. Note that such decomposition reduces the parametrization to a rank-1 matrix, $\boldsymbol{c}_{\boldsymbol{\theta}}(\boldsymbol{x})\boldsymbol{c}_{\boldsymbol{\theta}}(\boldsymbol{x})^T$ and a positive scalar, $\lambda$. Therefore, it gains computational efficiency by acknowledging the singular structure of $\mathbf{\Sigma}^{opt}$ while crucially remaining full rank for numerical stability [3].

**$\lambda$ and Loss Attenuation.** Interestingly, $\lambda$ affects the loss beyond stability. Considering a binary classification for simplicity, with $\lambda = 0$, the optimal variance per-example is a scalar $\sigma_i^{opt} = (y_i - \mu_i)^2$. Using the optimal variances in the label-dependent term of the loss in Equation 5 results in each term being 1, as $(y_i - \mu_i)^2/\sigma_i^{opt} = (y_i - \mu_i)^2/(y_i - \mu_i)^2 = 1$. Furthermore, the gradients for $\mu_i$ in Equation 8 become $\frac{\partial \mathcal{L}}{\partial \mu_i}|_{\sigma^{opt}} = -(y_i - \mu_i)/\sigma_i^{opt} = -(y_i - \mu_i)/(y_i - \mu_i)^2 = -1/(y_i - \mu_i)$. This, again, clearly shows the loss attenuation properties of the LN likelihood, i.e., it increases and reduces the contribution of low and high residual examples, respectively.

How does $\lambda$ affect this behaviour? It acts as a threshold (on residuals) for where loss attenuation occurs. To see this, we note that the minimum value for any $\sigma_{\boldsymbol{\theta}}$ is $\lambda^2$, as $\sigma_{\boldsymbol{\theta}} = \sigma_{\boldsymbol{\theta}}^{\frac{1}{2}}\sigma_{\boldsymbol{\theta}}^{\frac{1}{2}}$, and the minimum value for $\sigma_{\boldsymbol{\theta}}^{\frac{1}{2}}$ is $\lambda$, due to the parametrization in Equation 9. Thus, as $\sigma_i^{opt}$ cannot be smaller than $\lambda^2$, the desired $\sigma_i^{opt} = (y_i - \mu_i)^2$ is unattainable when $(y_i - \mu_i)^2 < \lambda^2$. Instead, the optimization gets as close as possible, resulting in $\sigma_i^{opt} = \lambda^2$ (details in Appendix B.4). In general, we have $\sigma_i^{opt} = \max\left(\lambda^2, (y_i - \mu_i)^2\right)$. Therefore, when $(y_i - \mu_i)^2 \leq \lambda^2$, the label-dependent loss becomes $(y_i - \mu_i)^2/\sigma_i^{opt} = (y_i - \mu_i)^2/\lambda^2$, and the gradients become $-(y_i - \mu_i)/\sigma_i^{opt} = -(y_i - \mu_i)/\lambda^2$. We show the effect of the loss attenuation threshold $\lambda$ in Figure 2.

---

[3]Simple and efficient implementations of normal distributions with low-rank covariance matrices can be done in, *e.g.*, the distribution packages of TensorFlow (Dillon et al., 2017) and PyTorch.

### 3.2 The Softmax Centered Function

**Softmax-Centered with Temperature $S_C^\tau$.** We incorporate a temperature $\tau$ in the softmax-centered function by seeing it as a bijective scale function:

$$S_C^\tau(\boldsymbol{z}) = S_C \circ \text{Scale}_{1/\tau}(\boldsymbol{z}) = S_C(\boldsymbol{z}/\tau), \tag{10}$$

$$(S_C^\tau)^{-1}(\boldsymbol{q}) = \text{Scale}_{1/\tau}^{-1} \circ S_C^{-1}(\boldsymbol{q}) = \tau S_C^{-1}(\boldsymbol{q}), \tag{11}$$

where $\boldsymbol{z} \in \mathbb{R}^{K-1}$ and $\boldsymbol{q} \in \mathring{\Delta}^{K-1}$. By introducing temperature in the softmax centered bijection, the target logit in the log-likelihood of Equation 5 changes from $\boldsymbol{y}_i$ to $\tau\boldsymbol{y}_i$, *i.e.*, $\tau$ controls the magnitude of the target logit vector. We believe this has two major effects on learning: i) a low $\tau$ moves the target logit closer to the origin, making it easier for the network to match $\boldsymbol{\mu}$ to it. Conversely, a large $\tau$ imposes a challenge as the network must output $\boldsymbol{\mu}$ with large magnitudes, which it is penalized from doing by the log prior (weight decay). ii) The chosen $\tau$ controls the range of residuals in the mean squared error loss. To illustrate, consider a sample with a noisy target logit $\tau\boldsymbol{y}_i$, for which the network predicts the true target $\boldsymbol{\mu} = \tau\boldsymbol{y} = \tau S_C^{-1}((1-t)\boldsymbol{\delta}_j + t\boldsymbol{u})$ with $y_i \neq j$. Then the negative log-likelihood for this example with an identity covariance is:

$$\begin{aligned}
&(\tau\boldsymbol{y}_i - \boldsymbol{\mu}_{\boldsymbol{\theta}}(\boldsymbol{x}_i))^T(\tau\boldsymbol{y}_i - \boldsymbol{\mu}_{\boldsymbol{\theta}}(\boldsymbol{x}_i)) \\
&= (\tau\boldsymbol{y}_i - \tau\boldsymbol{y})^T(\tau\boldsymbol{y}_i - \tau\boldsymbol{y}) \\
&= \tau^2(\boldsymbol{y}_i - \boldsymbol{y})^T(\boldsymbol{y}_i - \boldsymbol{y}) = \tau^2 C
\end{aligned} \tag{12}$$

where $C = (\boldsymbol{y}_i - \boldsymbol{y})^T(\boldsymbol{y}_i - \boldsymbol{y})$ is the loss without temperature scaling. Hence, the temperature determines how much the mean squared error part of the loss penalizes deviations from the observed target. In this work, we treat the temperature $\tau$ as a hyperparameter.

**Softmax Centered with a Dummy Class.** An issue with the softmax centered function is that it treats the last class differently from the rest. To see this, we look at the target logits ($\boldsymbol{y}$) for target categoricals ($S_C(\boldsymbol{y})$) for different classes ($y$), for details see Appendix B.5. If $y$ is not the last class, then $\boldsymbol{y}$ is zero in all components except for component $y$ where it is a constant ($\mathcal{C}$) that depends on the number of classes $K$. However, if $y$ is the last class, then all components of $\boldsymbol{y}$ are $-\mathcal{C}$. See Figure 1 (left) where $\boldsymbol{y}_1 = [\mathcal{C}, 0]$, $\boldsymbol{y}_2 = [0, \mathcal{C}]$ and $\boldsymbol{y}_3 = [-\mathcal{C}, -\mathcal{C}]$. This becomes a bigger problem for large $K$, as the squared l2-norm for an observed target logit is $(K-1)\mathcal{C}^2$ for the last class, and $\mathcal{C}^2$ otherwise.

As it is only the last class that is treated differently, we mitigate this by introducing a dummy class. We make $S_C(\boldsymbol{y})$ be in $\mathring{\Delta}^K$ by having the delta and uniform distribution be over $K + 1$ classes, and the network therefore has to output mean and covariance in $\mathbb{R}^K$ and $\mathbb{R}^{K \times K}$, respectively. Then, all the $K$ first classes that we care about are treated equally.

### 3.3 Predictions on Unseen Data

With our noise model in Equation 3, we relate the observed noisy target with the true target via an additive normally-distributed noise in the logit space. For unseen data $\boldsymbol{x}^*$, however, we would like to predict the true target, not the noisy target, and, therefore, we use $S_C(\boldsymbol{\mu}_{\boldsymbol{\theta}}(\boldsymbol{x}^*))$ instead, i.e., setting $\boldsymbol{\epsilon}(\boldsymbol{x}^*) = 0$. This means that we can discard the network's head predicting $\boldsymbol{\Sigma}$ after training.

## 4 Related Work

**Heteroscedastic Noise Estimation.** Nix & Weigend (1994) tackle the problem of input-dependent noise, for regression, in a maximum likelihood estimation framework. They assume additive Gaussian label noise and optimize two different networks to predict the mean and the variance of the output distributions. Kendall & Gal (2017) importantly note that such a framework provides a model with the capacity to effectively attenuate the loss induced by samples which are hard to model (Equation 2) and thus renders it possibly robust to label noise. Furthermore, they argue that such attenuation properties are also desirable in classification and propose a method termed Heteroscedastic Classification NNs (Het). They place a normal distribution over

the logits to model heteroscedastic noise, which is then marginalized out to obtain a categorical distribution:

$$\mathbb{E}_{\boldsymbol{\epsilon}(\boldsymbol{x}_i)\sim\mathcal{N}(\mathbf{0},\boldsymbol{\Sigma}_{\boldsymbol{\theta}}(\boldsymbol{x}_i))}[S(\boldsymbol{\mu}_{\boldsymbol{\theta}}(\boldsymbol{x}_i) + \boldsymbol{\epsilon}(\boldsymbol{x}_i))] \tag{13}$$

where $\boldsymbol{\mu} \in \mathbb{R}^K$, and $\boldsymbol{\Sigma} \in \mathbb{R}^{K \times K}$ is a diagonal covariance matrix, both predicted by a neural network, and $S$ is the standard softmax. The negative log-likelihood of this categorical distribution is then used as the loss. Collier et al. (2020) extended this by tempering the softmax (Het-$\tau$), and evaluated the robustness of the method to label noise. In another work, Collier et al. (2021) proposed an efficient low-rank parameterization for the covariance matrices (Het-$\tau$-$\boldsymbol{\Sigma}_{full}$), which is similar to ours in Equation 9. To better understand these works, we analyze the gradients w.r.t. $\boldsymbol{\mu}$ for sample $i$ with label $y_i = c$ (derivation in Appendix B.6):

$$-\left(\boldsymbol{\delta}_c - \mathbb{E}_{\boldsymbol{\epsilon}}\left[S(\boldsymbol{\mu}_i + \boldsymbol{\epsilon})\frac{S(\boldsymbol{\mu}_i + \boldsymbol{\epsilon})_c}{\mathbb{E}_{\boldsymbol{\epsilon}}[S(\boldsymbol{\mu}_i + \boldsymbol{\epsilon})_c]}\right]\right) \tag{14}$$

Comparing the expectation in Equation 14 with the expectation in Equation 13, we see it is modified by a scalar factor to increase the contribution of each sampled categorical distribution $S(\boldsymbol{\mu}_i + \boldsymbol{\epsilon})$ that has a high confidence in the given class, $S(\boldsymbol{\mu}_i + \boldsymbol{\epsilon})_c$. Clearly, this is doing loss attenuation as the network could learn to add noise to increase the confidence in the given class, making the expected categorical be closer to the target one-hot distribution. However, the loss attenuation of this method is different from the one in maximum likelihood estimation with Normal (regression) and Logistic-Normal (classification) likelihoods, see Section 2.4. We empirically compare Logistic Normal with all the variants in this line of work.

**Loss Correction.** Closest to our work, are the loss correction methods that estimate the true categorical distribution and transform it to the observed noisy one (Sukhbaatar et al., 2014; Patrini et al., 2017): $\tilde{p}(y|x) = \boldsymbol{T}p(y|x)$, where $\boldsymbol{T}$ is a matrix with elements $T_{ij}$ estimating the probability that the noisy class is $j$, given that the true class is $i$. This is clearly related to our noise model, as we transform the true categorical to the noisy one by adding noise in logit space. Importantly, we estimate a per-example covariance matrix, while these works estimate a single matrix $\boldsymbol{T}$ per dataset.

**Loss Reweighting.** These methods propose to weight the per-example losses to reduce the contribution of noisy examples. These weights can be estimated through density estimation (Liu & Tao, 2015), or predicted by the same network (Wang et al., 2017; Thulasidasan et al., 2019), another network (Jiang et al., 2018), or via meta-learning (Ren et al., 2018). These methods need to avoid the trivial solution of all weights being 0, which is typically done by engineering an extra regularization term. Our method's loss attenuation is a form of reweighting. However, in contrast, we naturally extend a classic noise model that leads to a likelihood, which combined with a standard MAP estimation framework, directly leads to our loss with inherent regularization. This makes our design choices more interpretable and more conducive to further extensions.

**Memorization Effects.** These methods rely on the observation that neural networks learn easy (correctly labeled) examples first (Arpit et al., 2017). Examples with small loss can therefore be assumed to be correctly labeled and selected for learning. Some notable methods making use of this are: MentorNet (Jiang et al., 2018) with predefined curriculums, the Co-teaching methods (Han et al., 2018; Yu et al., 2019), and DivideMix Li et al. (2020a). Incorporating similar small-loss tricks in our method could further improve robustness.

**Robust Loss Functions.** Ghosh et al. (2017) proved that, for certain (symmetric) loss functions, the globally optimal classifier is the same when trained with noise-free data as when trained with symmetric or asymmetric noise, under certain assumptions. Based on this theory, several new loss functions have been proposed (Zhang & Sabuncu, 2018; Ma et al., 2020; Englesson & Azizpour, 2021) and even extensions of the theory (Zhou et al., 2021). These theoretical works are commendable, however, most assume access to unlimited data, which makes it unclear how the results translate to standard finite classification datasets.

**Regularization.** Several standard regularization methods have also been studied when training with label noise, e.g., label smoothing (Lukasik et al., 2020), dropout (Rusiecki, 2020; Goel & Chen, 2021), and early stopping (Li et al., 2020b; Bai et al., 2021). Furthermore, some methods regularize the predictions of the network to be consistent with predictions from earlier in training (Liu et al., 2020; Laine & Aila, 2017), while others add noise to the gradients, *e.g.*, by adding noise to the one-hot labels (Chen et al., 2020). Although many of these methods only *implicitly* tackle the underlying problem of label noise via standard regularization techniques, they show impressive empirical robustness. We believe that targeted methods like ours that *explicitly* tackles the problem could be naturally combined with these methods.

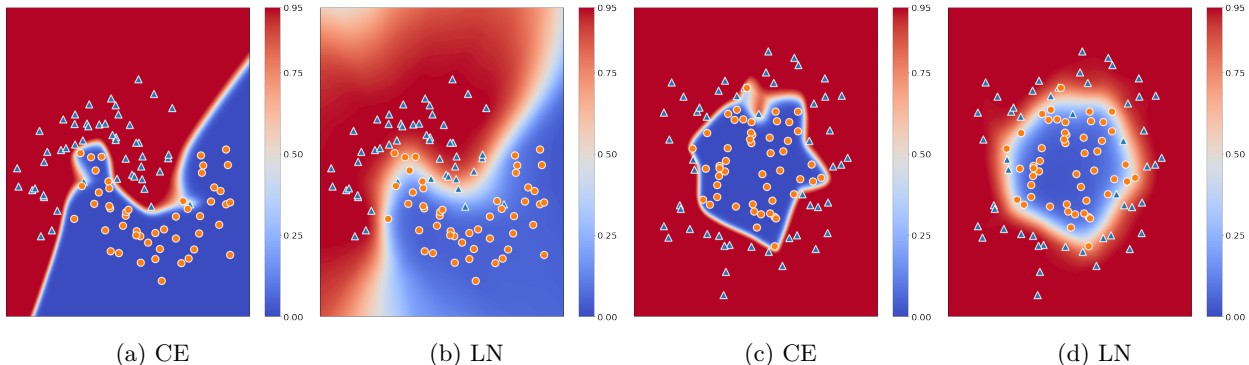

|  (a) CE | (b) LN | (c) CE | (d) LN |

Figure 3: **Synthetic Datasets.** To gain insights into the behavioral differences between our method (LN) and the cross-entropy (CE) loss, we study two binary datasets: the Two Moons dataset (left), and a dataset of two circles of different sizes (right). The color indicates the probability of the blue triangle class. Training with CE makes the network fit all the examples, resulting in a complex decision boundary. In contrast, the loss attenuation of our method reduces the need to fit all examples, leading to a smooth decision boundary.

## 5 Results

Here, we empirically verify that our theoretically motivated method demonstrates robustness to label noise. We describe the common training setup (Section 5.1) and the baselines (Section 5.2) and present results on synthetic datasets (Section 5.3), synthetic label noise (Section 5.4), and natural label noise (Section 5.5). Finally, we conduct additional insightful experiments studying different aspects of our method (Section 5.6).

### 5.1 Experimental Setup

We implement our method and the baselines in the same code base and compare on the following datasets: Two Moons & Circles, MNIST (Deng, 2012), CIFAR-10 & CIFAR-100 (Krizhevsky et al., 2009), CIFAR-10N & CIFAR-100N (Wei et al., 2022), and Clothing1M (Xiao et al., 2015). For all methods, and on all datasets, we search for method-specific hyperparameters based on noisy validation accuracy at the end of training. We report the mean and standard deviation of the test accuracy at the end of training for five different random seeds with the optimal hyperparameters. The seeds affect the network initialization, data loaders, and the generation of the synthetic label noise. For details about the experimental setup, see Appendix C.

### 5.2 Baselines

In addition to the standard cross-entropy (CE) loss, we compare our method with methods sharing the same motivation and goal in having a method that deals with heteroscedastic label noise similar to the probabilistic method for regression: Het (Kendall & Gal, 2017), Het$^\tau$ (Collier et al., 2020), and Het$^\tau_{\mathbf{\Sigma_{full}}}$ (Collier et al., 2021). Our method and these baselines model the pre-softmax logit vector as being normally distributed and use deep neural networks to output the mean and covariance of this distribution. We also compare with the loss correction method Forward (Patrini et al., 2017), and label smoothing (LS) regularization (Lukasik et al., 2020). Additionally, for completeness, we consider more distant baselines: Generalized Cross Entropy (GCE) (Zhang & Sabuncu, 2018), and Noise Against Noise (NAN) (Chen et al., 2020).

### 5.3 Synthetic Datasets: Two Moon & Circles

In Figure 3, we compare the behavior of our method (LN) with the cross entropy (CE) loss on two synthetic binary classification datasets: a dataset where the classes correspond to two half moons, and another with two circles of different radii. We find that training with the CE loss makes the network classify almost all examples according to their observed targets, resulting in a complex decision boundary that does not generalize well. In contrast, the network trained with the log-likelihood of the Logistic-Normal distribution has a smoother decision boundary, as the network is not classifying some examples as their given targets.

Table 1: **Synthetic Noise on MNIST, CIFAR-10 and CIFAR-100.** We implement our method and all baselines in the same shared code based and do a search for the best hyperparameters on a noisy validation set for all methods. We report the mean and standard deviation of the test accuracy from five runs with different random seeds. Our method (LN) shows strong performance compared to baselines, especially on the predictable asymmetric noise, and the challenging CIFAR-100 dataset.

| | Method | No Noise | Symmetric Noise Rate | | | Asymmetric Noise Rate | | |
| --- | --- | --- | --- | --- | --- | --- | --- | --- |
| | | 0% | 20% | 40% | 60% | 20% | 30% | 40% |
| MNIST | CE | $99.27 \pm 0.07$ | $88.41 \pm 0.34$ | $70.67 \pm 1.30$ | $51.04 \pm 1.19$ | $91.09 \pm 0.79$ | $86.31 \pm 1.25$ | $80.31 \pm 1.81$ |
| | GCE | $99.22 \pm 0.06$ | $\mathbf{98.85 \pm 0.18}$ | $\mathbf{98.60 \pm 0.11}$ | $\mathbf{97.45 \pm 0.31}$ | $98.52 \pm 0.31$ | $86.36 \pm 0.86$ | $79.81 \pm 1.46$ |
| | NAN | $98.44 \pm 0.24$ | $97.51 \pm 0.37$ | $90.03 \pm 0.95$ | $74.00 \pm 2.92$ | $96.46 \pm 2.23$ | $95.43 \pm 1.09$ | $88.95 \pm 1.63$ |
| | Forward | $99.27 \pm 0.03$ | $87.46 \pm 0.70$ | $69.96 \pm 2.10$ | $50.43 \pm 1.43$ | $91.95 \pm 0.40$ | $86.31 \pm 0.43$ | $80.97 \pm 1.23$ |
| | LS | $\mathbf{99.35 \pm 0.06}$ | $89.92 \pm 0.85$ | $69.07 \pm 0.93$ | $47.77 \pm 2.00$ | $92.06 \pm 0.95$ | $86.50 \pm 0.44$ | $80.00 \pm 0.93$ |
| | Het | $99.28 \pm 0.05$ | $87.09 \pm 0.70$ | $70.30 \pm 1.10$ | $50.57 \pm 1.02$ | $91.12 \pm 1.29$ | $86.35 \pm 0.62$ | $80.04 \pm 0.87$ |
| | $\text{Het}^{\tau}$ | $99.25 \pm 0.07$ | $88.10 \pm 0.70$ | $70.19 \pm 1.58$ | $51.95 \pm 1.13$ | $91.13 \pm 1.25$ | $85.95 \pm 0.69$ | $81.37 \pm 1.14$ |
| | $\text{Het}^{\tau}_{\mathbf{\Sigma_{full}}}$ | $\mathbf{99.25 \pm 0.15}$ | $89.16 \pm 0.44$ | $70.34 \pm 1.20$ | $50.89 \pm 1.79$ | $91.06 \pm 0.24$ | $86.25 \pm 0.88$ | $79.93 \pm 0.44$ |
| | LN | $\mathbf{99.38 \pm 0.06}$ | $98.53 \pm 0.27$ | $97.21 \pm 0.38$ | $90.93 \pm 2.29$ | $\mathbf{99.19 \pm 0.10}$ | $\mathbf{99.01 \pm 0.19}$ | $\mathbf{96.54 \pm 1.20}$ |
| CIFAR-10 | CE | $\mathbf{90.67 \pm 0.80}$ | $73.54 \pm 1.01$ | $56.56 \pm 1.44$ | $39.44 \pm 1.87$ | $81.35 \pm 1.26$ | $76.01 \pm 2.67$ | $71.89 \pm 1.67$ |
| | GCE | $\mathbf{90.83 \pm 0.44}$ | $\mathbf{87.55 \pm 0.41}$ | $\mathbf{84.72 \pm 0.82}$ | $\mathbf{64.28 \pm 1.42}$ | $85.68 \pm 0.69$ | $83.97 \pm 0.52$ | $72.90 \pm 1.61$ |
| | NAN | $89.61 \pm 0.93$ | $83.86 \pm 1.03$ | $79.80 \pm 0.59$ | $73.58 \pm 0.41$ | $84.32 \pm 1.05$ | $76.79 \pm 2.28$ | $72.90 \pm 1.92$ |
| | Forward | $\mathbf{90.69 \pm 0.38}$ | $74.39 \pm 1.49$ | $59.60 \pm 1.40$ | $40.06 \pm 2.16$ | $82.10 \pm 1.09$ | $77.02 \pm 2.38$ | $72.77 \pm 1.43$ |
| | LS | $89.78 \pm 0.39$ | $79.09 \pm 0.96$ | $64.27 \pm 1.50$ | $43.57 \pm 3.13$ | $81.99 \pm 1.22$ | $76.49 \pm 1.17$ | $71.66 \pm 1.78$ |
| | Het | $\mathbf{90.41 \pm 0.69}$ | $74.67 \pm 1.06$ | $58.53 \pm 1.96$ | $39.51 \pm 2.53$ | $81.72 \pm 1.65$ | $76.97 \pm 1.17$ | $72.88 \pm 1.53$ |
| | $\text{Het}^{\tau}$ | $\mathbf{91.18 \pm 0.41}$ | $76.90 \pm 1.79$ | $63.55 \pm 2.27$ | $44.73 \pm 1.67$ | $81.40 \pm 0.96$ | $77.41 \pm 2.53$ | $72.53 \pm 1.83$ |
| | $\text{Het}^{\tau}_{\mathbf{\Sigma_{full}}}$ | $\mathbf{90.82 \pm 0.42}$ | $77.16 \pm 0.94$ | $62.85 \pm 1.88$ | $44.20 \pm 3.05$ | $81.55 \pm 0.80$ | $77.05 \pm 0.33$ | $72.69 \pm 0.89$ |
| | LN | $90.17 \pm 0.55$ | $86.13 \pm 1.03$ | $81.37 \pm 1.97$ | $76.08 \pm 0.63$ | $\mathbf{87.64 \pm 0.78}$ | $\mathbf{86.91 \pm 1.03}$ | $\mathbf{82.18 \pm 1.30}$ |
| CIFAR-100 | CE | $\mathbf{64.87 \pm 0.88}$ | $47.39 \pm 0.43$ | $33.62 \pm 0.79$ | $20.04 \pm 0.58$ | $50.98 \pm 0.88$ | $44.04 \pm 0.73$ | $36.95 \pm 0.58$ |
| | GCE | $\mathbf{64.33 \pm 0.83}$ | $\mathbf{61.67 \pm 0.67}$ | $\mathbf{53.96 \pm 1.40}$ | $42.85 \pm 0.79$ | $59.63 \pm 1.28$ | $49.21 \pm 0.53$ | $36.78 \pm 0.50$ |
| | NAN | $\mathbf{64.25 \pm 0.64}$ | $56.93 \pm 0.77$ | $50.03 \pm 0.62$ | $40.45 \pm 0.41$ | $56.40 \pm 1.07$ | $52.78 \pm 0.85$ | $40.59 \pm 0.84$ |
| | Forward | $\mathbf{64.33 \pm 0.73}$ | $47.90 \pm 0.93$ | $32.28 \pm 1.10$ | $20.00 \pm 0.75$ | $50.82 \pm 0.57$ | $43.87 \pm 0.47$ | $37.02 \pm 0.72$ |
| | LS | $\mathbf{65.39 \pm 0.40}$ | $57.08 \pm 0.70$ | $44.03 \pm 1.20$ | $26.13 \pm 1.45$ | $55.47 \pm 0.76$ | $44.70 \pm 0.73$ | $38.56 \pm 0.66$ |
| | Het | $\mathbf{64.48 \pm 0.31}$ | $48.40 \pm 1.32$ | $34.26 \pm 0.37$ | $20.33 \pm 0.31$ | $51.44 \pm 1.15$ | $45.09 \pm 0.40$ | $37.43 \pm 0.66$ |
| | $\text{Het}^{\tau}$ | $\mathbf{64.20 \pm 0.37}$ | $54.17 \pm 0.79$ | $42.03 \pm 0.84$ | $22.33 \pm 0.57$ | $59.89 \pm 0.54$ | $53.75 \pm 1.08$ | $41.14 \pm 0.98$ |
| | $\text{Het}^{\tau}_{\mathbf{\Sigma_{full}}}$ | $\mathbf{65.18 \pm 0.90}$ | $54.83 \pm 0.46$ | $41.49 \pm 1.53$ | $22.42 \pm 0.95$ | $61.29 \pm 0.46$ | $56.44 \pm 0.53$ | $45.75 \pm 1.02$ |
| | LN | $\mathbf{64.88 \pm 0.98}$ | $\mathbf{60.58 \pm 1.07}$ | $\mathbf{55.55 \pm 1.30}$ | $\mathbf{46.43 \pm 1.15}$ | $\mathbf{64.31 \pm 0.98}$ | $\mathbf{64.07 \pm 0.77}$ | $\mathbf{61.20 \pm 1.22}$ |

## 5.4 Synthetic Noise

**Noise types.** Here, we study our method on two types of synthetic class-dependent label noise (asymmetric and symmetric). For each training sample, there is a risk (according to the noise rate) that its label is randomly re-sampled from a uniform distribution over the classes (symmetric noise) or changed to another, often perceptually similar, class (asymmetric). The asymmetric noise changes the labels as follows: MNIST: $7 \rightarrow 1, 2 \rightarrow 7, 5 \leftrightarrow 6, 3 \rightarrow 8$, CIFAR-10: bird $\rightarrow$ airplane, cat $\leftrightarrow$ dog, deer $\rightarrow$ horse, CIFAR-100: cyclically to the next class, *e.g.*, $1 \rightarrow 2$ and $99 \rightarrow 0$.

**Results.** Table 1 shows the results using symmetric and asymmetric label noise. Compared to the most related works (Forward and Het methods), we find that the test accuracy of our method is degraded the least for all noise types and rates. Out of the more general set of baselines, we find that the robust GCE loss performs remarkably well on symmetric noise. Our method shows largest improvements in robustness for the more challenging CIFAR-100 dataset, as well as for asymmetric noise. For example, on CIFAR-100 with 40% asymmetric noise, our method achieves a mean test error of ∼61% compared to ∼46% of the best baseline. We find that the generalization of the networks trained with our method is barely affected when increasing asymmetric noise rates from 20% to 30% on all datasets. This is likely due to the predictable structure of the

Table 2: **Natural Noise: CIFAR-10N, CIFAR-100N, and Clothing1M.** For all methods, we search for method-specific hyperparameters based on a noisy validation set and report the mean and standard deviation for the best setting. Our method performs as well as other baselines, or is a close second to the robust GCE (Zhang & Sabuncu, 2018) loss.

| Method | CIFAR-10N | | | | | CIFAR-100N | Clothing1M |
|---|---|---|---|---|---|---|---|
| | Random 1 | Random 2 | Random 3 | Aggregate | Worst | | |
| CE | $77.75 \pm 0.74$ | $75.52 \pm 1.08$ | $76.25 \pm 1.26$ | $83.59 \pm 0.98$ | $59.01 \pm 0.98$ | $42.75 \pm 0.93$ | $71.04 \pm 0.15$ |
| GCE | $\mathbf{85.66 \pm 0.73}$ | $\mathbf{85.58 \pm 0.65}$ | $\mathbf{84.78 \pm 0.62}$ | $\mathbf{86.66 \pm 0.68}$ | $\mathbf{77.48 \pm 1.22}$ | $48.81 \pm 0.46$ | $71.95 \pm 0.21$ |
| NAN | $81.85 \pm 1.13$ | $83.40 \pm 0.84$ | $82.77 \pm 0.78$ | $85.53 \pm 0.83$ | $75.47 \pm 0.76$ | $\mathbf{50.00 \pm 0.72}$ | $71.50 \pm 0.41$ |
| Forward | $77.97 \pm 0.80$ | $77.13 \pm 0.74$ | $77.51 \pm 1.21$ | $83.58 \pm 1.50$ | $58.91 \pm 0.57$ | $42.53 \pm 0.41$ | $70.75 \pm 0.25$ |
| LS | $80.07 \pm 0.82$ | $79.81 \pm 0.62$ | $79.41 \pm 0.68$ | $85.08 \pm 0.54$ | $63.07 \pm 1.93$ | $45.98 \pm 1.44$ | $\mathbf{72.04 \pm 0.42}$ |
| Het | $76.38 \pm 0.97$ | $75.85 \pm 1.45$ | $76.18 \pm 1.60$ | $83.87 \pm 1.02$ | $58.86 \pm 1.27$ | $42.90 \pm 0.48$ | $70.87 \pm 0.38$ |
| $\mathrm{Het}^\tau$ | $78.83 \pm 1.65$ | $78.29 \pm 1.61$ | $78.27 \pm 0.86$ | $84.34 \pm 0.48$ | $62.01 \pm 2.03$ | $45.82 \pm 0.53$ | $\mathbf{72.24 \pm 0.30}$ |
| $\mathrm{Het}^\tau_{\mathbf{\Sigma_{full}}}$ | $78.87 \pm 0.47$ | $76.24 \pm 0.96$ | $77.68 \pm 1.93$ | $84.45 \pm 0.57$ | $63.27 \pm 2.62$ | $45.58 \pm 0.80$ | $\mathbf{72.41 \pm 0.15}$ |
| LN | $83.70 \pm 0.80$ | $83.65 \pm 0.82$ | $\mathbf{84.07 \pm 0.71}$ | $\mathbf{85.35 \pm 1.33}$ | $74.31 \pm 1.08$ | $\mathbf{50.37 \pm 0.50}$ | $72.03 \pm 0.52$ |

asymmetric noise, which could also explain why our method fall behind the robust GCE loss on symmetric noise. The predictability of the noise is important, as we use a neural network to predict the noise variance of the LN likelihood. As the optimal $\mathbf{\Sigma}$ in Equation 7 depend on the residual, which in turn depend on the label, the network needs to predict the noisy label for the mislabeled examples. However, for uniform noise, the labels are inherently unpredictable, and therefore the network has to resort to memorization.

### 5.5 Natural Noise

**Datasets.** Synthetic label noise is excellent for studying robustness to noise under controlled noise rates. However, this comes at the cost of the structure of the noise (class-dependent) potentially being different from what one would observe in practice (input-dependent), *e.g.*, due to mistakes in the annotation process. In this section, we study the robustness of our method on natural noise by using the recently proposed CIFAR-N datasets (Wei et al., 2022) and Clothing1M (Xiao et al., 2015), see Appendix D for more information.

**Results.** Table 2 shows the test accuracy of our method on naturally noisy datasets. Compared to the most relevant baselines (Forward and Het methods), we find that networks trained with our method generalize better on the CIFAR-N datasets, and as good as $\mathrm{Het}^\tau$ and $\mathrm{Het}^\tau_{\mathbf{\Sigma_{full}}}$ on Clothing1M. For the more general set of baselines, NAN, and especially GCE, have strong performance in this setting.

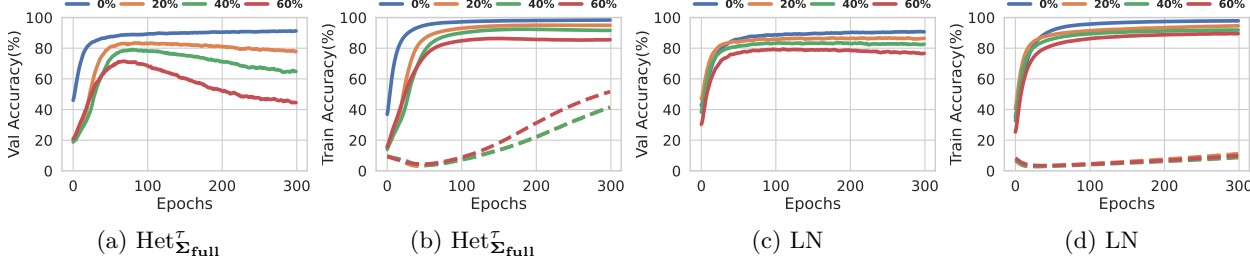

(a) $\mathrm{Het}^\tau_{\mathbf{\Sigma_{full}}}$      (b) $\mathrm{Het}^\tau_{\mathbf{\Sigma_{full}}}$      (c) LN      (d) LN

Figure 4: **The Evolution of Validation and Training Accuracy.** We plot the noise-free validation accuracy and noisy training accuracy of $\mathrm{Het}^\tau_{\mathbf{\Sigma_{full}}}$ (left) and our method (right) on varying symmetric noise rates during training on CIFAR-10. We report the training accuracy of noise-free (full) and noisy (dashed) examples separately. We observe that the generalization degrades (a) for networks trained with $\mathrm{Het}^\tau_{\mathbf{\Sigma_{full}}}$ as it fits the noisy examples (b). Our method results in better generalization (c) and is more robust against fitting the noisy examples of the training set (d).

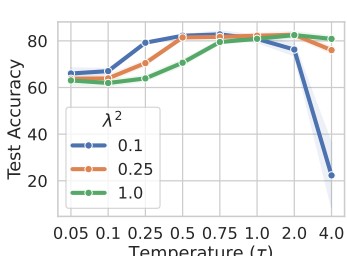
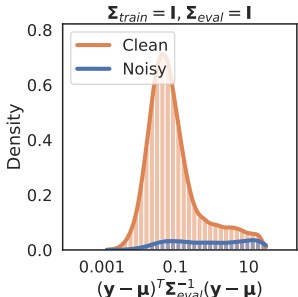
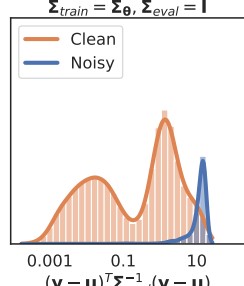
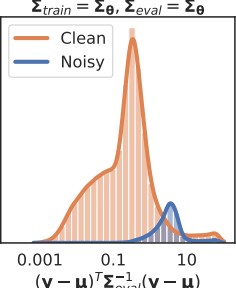

Figure 5: **Sensitivity to Hyperparameters.** The mean and standard deviation of test accuracy for various hyperparameters on CIFAR-10 with 40% symmetric noise. A too small $\tau$ leads to overfitting to noise, while a too large $\tau$ and $\lambda$ leads to slow convergence.

Figure 6: **Histogram of Residuals.** We train two LN networks with different covariance matrices: $\mathbf{\Sigma}_{train} = \boldsymbol{I}$, and $\mathbf{\Sigma}_{train} = \mathbf{\Sigma}_{\boldsymbol{\theta}}$. At the end of training, we calculate the distributions of the label-dependent term of the loss, $(\boldsymbol{y} - \boldsymbol{\mu})^T \mathbf{\Sigma}_{eval}^{-1} (\boldsymbol{y} - \boldsymbol{\mu})$, on the CIFAR-10 training set with 20% asymmetric noise. We use $\mathbf{\Sigma}_{train} = \mathbf{\Sigma}_{eval} = \boldsymbol{I}$ (left), and for the network with $\mathbf{\Sigma}_{train} = \mathbf{\Sigma}_{\boldsymbol{\theta}}$, we use $\mathbf{\Sigma}_{eval} = \boldsymbol{I}$ (middle), and $\mathbf{\Sigma}_{eval} = \mathbf{\Sigma}_{\boldsymbol{\theta}}$ (right). We find that $\boldsymbol{\mu}$ fits less of the noisy examples when learning $\mathbf{\Sigma}$ (cf. left and middle), and that $\mathbf{\Sigma}_{\boldsymbol{\theta}}$ increases and decreases the contribution of low- and high-residual examples, respectively (cf. middle and right).

## 5.6 Empirical Study of LN

**How does the accuracy evolve during training?** Figure 4 shows the clean validation and noisy training accuracy (for clean and noisy examples separately) on CIFAR-10 with symmetric noise for various noise rates for $\text{Het}_{\mathbf{\Sigma}_{\mathbf{full}}}^{\tau}$ and our method. The generalization of networks trained with $\text{Het}_{\mathbf{\Sigma}_{\mathbf{full}}}^{\tau}$ improves early in training, and degrades when the networks start fitting the noisy examples. In contrast, we find that networks trained with LN fit significantly fewer noisy examples and thus show a smaller decrease in generalization.

**How sensitive is LN to hyperparameters?** See Figure 5. A small $\tau$ makes the network overfit, likely due to the target logit being close to the origin (Equation 11), thus easy to fit. A large $\tau$ with a small $\lambda$ leads to slow convergence, likely due to some clean examples being loss attenuated as $\tau$ increases residuals (Equation 12), making examples more likely to be above the loss attenuation threshold, see last paragraph in Section 3.1.

**How is $\mathbf{\Sigma}_{\boldsymbol{\theta}}(\boldsymbol{x})$ affecting the network?** In Figure 6, a network trained with an identity matrix has, as expected, more noisy examples with lower residuals (left), compared to the network that learns the covariance matrix (middle). Interestingly, the residuals in the middle figure are bimodal, which we believe is due to some clean examples being below the learnable loss attenuation threshold (Section 3.1) and some above. Comparing Figure 6 middle and right, we find that $\mathbf{\Sigma}_{\boldsymbol{\theta}}$ is increasing the residuals of some clean examples, while also reducing the residuals of noisy ones mixed with some (hard) clean samples. These results are for the CIFAR-10 training set with 20% asymmetric noise, see Appendix G.3 for more noise types.

**How important is learning a full $\mathbf{\Sigma}_{\boldsymbol{\theta}}(\boldsymbol{x})$?** In Table 3, we train with LN on CIFAR-10N (noise type "worst") and CIFAR-100N with different per-example covariance matrices: $\boldsymbol{I}$ (Identity), $\sigma^2 \boldsymbol{I}$ (Isotropic), $\text{diag}([\sigma_1^2, \ldots, \sigma_K^2])$ (Diag), and the parametrization in Equation 9 (Full). We observe that using an identity matrix generalizes significantly worse than all the other learnable ones, highlighting the importance of loss attenuation. Furthermore, we find our proposed parametrization (Full) to significantly outperform the rest.

Table 3: **Ablation Study.** We analyze the effect of learning $\mathbf{\Sigma}$, different parameterizations, and the dummy class. Learning a full $\mathbf{\Sigma}$ performs the best, and the dummy class is crucial for datasets with many classes.

| $\mathbf{\Sigma}$ | Dummy Class | CIFAR-10N | CIFAR-100N |
|---|---|---|---|
| Identity | ✓ | $65.64 \pm 1.53$ | $47.69 \pm 1.46$ |
| Isotropic | ✓ | $71.12 \pm 1.85$ | $46.44 \pm 0.67$ |
| Diagonal | ✓ | $69.87 \pm 1.11$ | $46.98 \pm 0.89$ |
| Full | ✓ | $\mathbf{74.31 \pm 1.08}$ | $\mathbf{50.37 \pm 0.50}$ |
| Full | ✗ | $\mathbf{74.72 \pm 1.14}$ | $26.31 \pm 1.43$ |

**How important is the dummy class?** Our analysis in Section 3.2 highlighted that the softmax centered treated the last class differently, and that this difference increased with the number of classes. In Table 3, we evaluate the importance of our solution (dummy class) to this problem. As expected, on CIFAR-10, it makes no significant difference, however, on CIFAR-100 with its many more classes, it becomes crucial.

## 6    Limitations and Future Work

Interleaved in the previous sections, we have addressed or discussed limitations of our work, *e.g.*, asymmetry of softmax centered → dummy class, full covariance matrices do not scale to high number of classes → low-rank approximation, the Logistic-Normal distribution is not defined on the borders of the probability simplex → label smoothing. Although our method is conceptually simple, these minor designs to make the extension work in practice are not surprising to us. We believe it could explain why such natural extension from regression has not happened before despite the establishment of the regression approach and the more practicality and wide applicability of the classification settings, especially for modern deep networks. Furthermore, in Section 5.4, we discuss the limitation of using neural networks to predict the noise variance in the LN likelihood. We believe an interesting future direction is improving this estimation. Next, we discuss more future work.

**Soft Labels**. We model the observed target as the softmax of a logit vector, $S_C(\boldsymbol{y})$. However, this leads to a limitation that the target cannot be on the borders of the probability simplex. We proposed label smoothing as a simple solution for this issue. However, as future work, there are many other interesting possibilities of obtaining a soft label, e.g., using the categorical prediction of another network, as in Knowledge Distillation (Hinton et al., 2014), or temporal ensembling (Laine & Aila, 2017), etc.

**Neural Network Point Estimates.** We assume the observed target logit can be modelled as the true target with additive noise in logit space: $\boldsymbol{y} = \boldsymbol{\mu} + \boldsymbol{\epsilon}, \boldsymbol{\epsilon} \sim \mathcal{N}(0, \boldsymbol{\Sigma})$. That is, we explain the residual $\boldsymbol{y} - \boldsymbol{\mu}$, as zero-mean normally distributed noise $\boldsymbol{\epsilon}$. Hence, high-residual examples will be attenuated. In this work, as $\boldsymbol{\mu}$ and $\boldsymbol{\Sigma}$ are unknown, we estimate them with a neural network. Therefore, a limitation of our work is that if estimates of $\boldsymbol{\mu}$ are poor, then we will incorrectly model the residual as noise, which could lead to *e.g.*, slower convergence. To improve these estimates, we believe exciting directions for future work are to extend our method to incorporate epistemic uncertainty and/or distance awareness, which has been done for the most related work to us (Het) by Kendall & Gal (2017) and Fortuin et al. (2022), respectively.

**Gaussian Process Classification.** Due to the duality between the logit space and the probability simplex, see Figure 1, we can interpret our method as: i) converting the classification labels to regression labels $(S_C^{-1}(\tilde{p}(y|\boldsymbol{x}_i)))$, ii) training a regression neural network with a Gaussian likelihood loss, and iii) making classification predictions on unseen examples by turning the regression predictions ($\boldsymbol{\mu}$) to categorical distributions ($S_C(\boldsymbol{\mu})$) via the softmax centered bijection. Gaussian Processes (GPs) for classification typically use the categorical likelihood, which requires approximate methods to find the approximate posterior predictive distributions. However, given the interpretation of our method above, we can get closed-form predictive posterior distributions even in classification. Similarly to above, the procedure would be: i) turn a classification dataset into a regression dataset using the softmax centered bijection, ii) train a regression GP on this dataset, iii) get closed-form posterior predictive distributions (Normal distributions) from the GP, and finally transform these predictions to classification predictions (Logistic-Normal distributions) by applying the softmax centered function. We expect this method to be much faster than the approximate methods, and it would be interesting to compare the quality of the predictions, especially in settings with label noise.

## 7    Conclusion

The goal of this work was to extend the simple and probabilistic approach of doing loss attenuation in regression to classification. We successfully achieved this by proposing a noise model that lead to the Logistic-Normal distribution. We proposed to learn the parameters of the distribution with neural networks through maximum likelihood estimation and formally presented the loss attenuation effects obtained when optimizing such models. Finally, we empirically verified that LN is effectively robust to label noise. As our method has the same loss attenuation as in the regression case, it can serve as a simple alternative to the methods of Kendall & Gal (2017); Collier et al. (2020; 2021).

**Acknowledgement**. This work was partially supported by the Wallenberg AI, Autonomous Systems and Software Program (WASP) funded by the Knut and Alice Wallenberg Foundation. All experiments were performed using the supercomputing resource Berzelius provided by the National Supercomputer Centre at Linköping University and the Knut and Alice Wallenberg foundation.

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

## A  Appendix

In the Appendix, we provide derivations of important theoretical results (Section B), describe details regarding hyperparameters (Section C) and the natural datasets (Section D). Furthermore, we provide implementation details of the method (Section E), discuss additional related work (Section F) and present additional experiments (Section G).

## B  Derivations

### B.1  The Probability Density Function of a Logistic-Normal Distribution

Assume $\mathbf{Y} \sim \mathcal{N}(\boldsymbol{\mu}, \boldsymbol{\Sigma})$ for some $\boldsymbol{\mu} \in \mathbb{R}^{K-1}, \boldsymbol{\Sigma} \in \mathbb{R}^{(K-1) \times (K-1)}$ and $\mathbf{S} = S_C(\mathbf{Y})$, then the probability density function (pdf) of $\mathbf{S}$ can be written as $f_{\mathbf{S}}(\boldsymbol{s}) = \text{abs}(|J(\boldsymbol{s})|) \cdot f_{\mathbf{Y}}(S_C^{-1}(\boldsymbol{s}))$. In which, $|J(\boldsymbol{s})|$, the determinant of the Jacobian of $S_C^{-1}(\boldsymbol{s}) = (\log \frac{\boldsymbol{s}_k}{\boldsymbol{s}_K})_{k \in \mathbb{N}_{K-1}}$ is given by Dillon et al. (2017)

$$
\begin{aligned}
|J(\boldsymbol{s})| &= |\mathbf{D}^{-1} + \frac{1}{s_K}| \\
&= |\mathbf{D}^{-1} + \frac{\mathbf{D}^{-1} \boldsymbol{s}_{-K} \boldsymbol{s}_{-K}^T \mathbf{D}^{-1}}{s_K}| \\
&= |\mathbf{D}^{-1} + \frac{\mathbf{D}^{-1} \boldsymbol{s}_{-K} \boldsymbol{s}_{-K}^T \mathbf{D}^{-1}}{1 - \boldsymbol{s}_{-K}^T \mathbf{D}^{-1} \boldsymbol{s}_{-K}}| \\
&= \left(|\mathbf{D} - \boldsymbol{s}_{-K} \boldsymbol{s}_{-K}^T|\right)^{-1} \\
&= \left(|\mathbf{D}|(1 - \boldsymbol{s}_{-K}^T \mathbf{D}^{-1} \boldsymbol{s}_{-K})\right)^{-1} \\
&= \left(\prod_{k=1}^{K} \boldsymbol{s}_k\right)^{-1}
\end{aligned}
\tag{15}
$$

where $\mathbf{D} = \text{diag}(\boldsymbol{s}_{-K})$ and $\boldsymbol{s}_{-K} = (\boldsymbol{s}_k)_{k \in \mathbb{N}_{K-1}}$. The first equality is obtained by computing derivatives while $\boldsymbol{s}_K = 1 - \sum_{k=1}^{K-1} \boldsymbol{s}_k$, the fourth equality is based on the Sherman–Morrison identity Sherman & Morrison (1950) and the fifth equality holds based on the matrix determinant lemma Harville (1998). As $f_{\mathbf{S}}(\boldsymbol{s}) = \text{abs}(|J(\boldsymbol{s})|) \cdot f_{\mathbf{Y}}(S_C^{-1}(\boldsymbol{s}))$, the pdf of the Logistic-Normal distribution is

$$
f_{\mathbf{S}}(\boldsymbol{s}) = \frac{1}{\prod_{k=1}^{K} \boldsymbol{s}_k} \frac{1}{|(2\pi)^{K-1} \boldsymbol{\Sigma}|^{\frac{1}{2}}} e^{-\frac{1}{2} \boldsymbol{r}^T \boldsymbol{\Sigma}^{-1} \boldsymbol{r}} = \frac{1}{\prod_{k=1}^{K} S_C(\boldsymbol{y})_k} \frac{1}{|(2\pi)^{K-1} \boldsymbol{\Sigma}|^{\frac{1}{2}}} e^{-\frac{1}{2} \boldsymbol{r}^T \boldsymbol{\Sigma}^{-1} \boldsymbol{r}}
\tag{16}
$$

where $\boldsymbol{r} = S_C^{-1}(\boldsymbol{s}) - \boldsymbol{\mu} = \boldsymbol{y} - \boldsymbol{\mu}$.

### B.2  Optimal Covariance Matrix

We want to show that the optimal $\boldsymbol{\Sigma}$ matrix for example $j$ is: $\boldsymbol{\Sigma}_j^{opt} = (\boldsymbol{y}_j - \boldsymbol{\mu}_j)(\boldsymbol{y}_j - \boldsymbol{\mu}_j)^T = \boldsymbol{r}_j \boldsymbol{r}_j^T$. First, we note that the gradients of the negative log likelihood in Equation 5 with respect to $\boldsymbol{\Sigma}_j$ is:

$$
\frac{\partial \mathcal{L}}{\partial \boldsymbol{\Sigma}_j} = \frac{1}{2} \sum_{i=1}^{N} \frac{\boldsymbol{r}_i^T \boldsymbol{\Sigma}_i^{-1} \boldsymbol{r}_i + \log |\boldsymbol{\Sigma}_i|}{\partial \boldsymbol{\Sigma}_j} = \frac{1}{2} \frac{\boldsymbol{r}_j^T \boldsymbol{\Sigma}_j^{-1} \boldsymbol{r}_j + \log |\boldsymbol{\Sigma}_j|}{\partial \boldsymbol{\Sigma}_j}
$$

as all other terms of the loss are unaffected by $\boldsymbol{\Sigma}_j$ and are therefore zero. Computing the gradients with respect to $\boldsymbol{\Sigma}_j$ while applying two identities $\frac{\partial}{\partial \boldsymbol{\Sigma}} \log |\boldsymbol{\Sigma}| = \boldsymbol{\Sigma}^{-1}$ and $\frac{\partial}{\partial \boldsymbol{\Sigma}} \boldsymbol{r}^T \boldsymbol{\Sigma}^{-1} \boldsymbol{r} = -\boldsymbol{\Sigma}^{-1} \boldsymbol{r} \boldsymbol{r}^T \boldsymbol{\Sigma}^{-1}$ (see Petersen et al. (2008) Equations 57 and 63), we have

$$
\frac{\partial}{\partial \boldsymbol{\Sigma}} [\boldsymbol{r}_j^T \boldsymbol{\Sigma}_j^{-1} \boldsymbol{r}_j + \log |\boldsymbol{\Sigma}_j|] = -\boldsymbol{\Sigma}_j^{-1} \boldsymbol{r}_j \boldsymbol{r}_j^T \boldsymbol{\Sigma}_j^{-1} + \boldsymbol{\Sigma}_j^{-1}
\tag{17}
$$

where the latter identity being justified by properties of symmetry and trace of matrices as follows

$$\frac{\partial}{\partial \boldsymbol{\Sigma}} \boldsymbol{r}^T \boldsymbol{\Sigma}^{-1} \boldsymbol{r} = \frac{\partial}{\partial \boldsymbol{\Sigma}} \text{tr}[\boldsymbol{r} \boldsymbol{r}^T \boldsymbol{\Sigma}^{-1}] = -(\boldsymbol{\Sigma}^{-1} \boldsymbol{r} \boldsymbol{r}^T \boldsymbol{\Sigma}^{-1})^T = -\boldsymbol{\Sigma}^{-1} \boldsymbol{r} \boldsymbol{r}^T \boldsymbol{\Sigma}^{-1} \tag{18}$$

Setting the term in the right-hand side of Equation 17 to zero then left and right multiplying by $\boldsymbol{\Sigma}_j$, yields the maximizer of the likelihood $\boldsymbol{\Sigma}_j^{opt} = \boldsymbol{r}_j \boldsymbol{r}_j^T$.

### B.3 Gradients for the Mean with Optimal Covariance

The optimal covariance $\boldsymbol{\Sigma}_j^{opt} = (\boldsymbol{y}_j - \boldsymbol{\mu}_j)(\boldsymbol{y}_j - \boldsymbol{\mu}_j)^T = \boldsymbol{r}_j \boldsymbol{r}_j^T$ is rank 1 and is therefore not invertible. However, for an invertible $\boldsymbol{\Sigma}_j$, we have

$$\frac{\partial \mathcal{L}}{\partial \boldsymbol{\mu}_j} = -\boldsymbol{\Sigma}_j^{-1} \boldsymbol{r}_j \Leftrightarrow \boldsymbol{\Sigma}_j \frac{\partial \mathcal{L}}{\partial \boldsymbol{\mu}_j} = -\boldsymbol{\Sigma}_j \boldsymbol{\Sigma}_j^{-1} \boldsymbol{r}_j = \boldsymbol{r}_j \tag{19}$$

This is a linear system of the form $\boldsymbol{A}\boldsymbol{x} = \boldsymbol{b}$, which can be solved with exact (or approximate) methods. If we assume $\boldsymbol{\Sigma}^{opt}$ is invertible (in practice this is done by adding a diagonal matrix), and solve the linear system with $\boldsymbol{A} = \boldsymbol{\Sigma}_j^{opt}$ instead, one solution is $\frac{\partial \mathcal{L}}{\partial \boldsymbol{\mu}_j} = \frac{\boldsymbol{r}_j}{||\boldsymbol{r}_j||_2^2}$ as $\boldsymbol{r}_j$ is an eigenvector of $\boldsymbol{\Sigma}_j^{opt}$ with eigenvalue $||\boldsymbol{r}_j||_2^2$:

$$\boldsymbol{\Sigma}_j^{opt} \boldsymbol{r}_j = (\boldsymbol{r}_j \boldsymbol{r}_j^T) \boldsymbol{r}_j = \boldsymbol{r}_j (\boldsymbol{r}_j^T \boldsymbol{r}_j) = ||\boldsymbol{r}_j||_2^2 \boldsymbol{r}_j \tag{20}$$

**Side-note**: In practice, this is not how the gradients are computed. Typically the label-dependent part of the loss is computed with a similar rewrite as above, i.e., $\boldsymbol{r}_j \boldsymbol{\Sigma}^{-1} \boldsymbol{r} = \boldsymbol{r}_j (\boldsymbol{\Sigma} \setminus \boldsymbol{r})$ where $\boldsymbol{\Sigma} \setminus \boldsymbol{r}$ is the (exact or approximate) solution to the linear system $\boldsymbol{\Sigma}\boldsymbol{x} = \boldsymbol{r}$, which we then do backpropagation through. This is more numerically stable than calculating the inverse and multiplying it with $\boldsymbol{r}_j$.

### B.4 The Effect of Lambda on the Optimal Variance

In this section, we provide derivations for the behavior in Figure 2. Our goal is to show how $\lambda$ affects the optimally learned $\sigma^2$. First, we note that Equation 5 for binary classification becomes

$$\mathcal{L} = \frac{1}{2} \sum_{i=1}^{N} \frac{(y_i - \mu_i)^2}{\sigma_i^2} + \log \sigma_i^2 + C \tag{21}$$

and we want to look at $\frac{\mathcal{L}}{\partial \sigma_i^2} = 0$ for a particular example $i$. To simplify notation, we let $\sigma_i^2 = \sigma^2$, $y_i = y$, and $\mu_i = \mu$ and let $r = y - \mu$ denote the residual. With this notation, the gradient of the loss with respect to $\sigma^2$ for example $i$ is

$$\frac{\partial}{\partial \sigma^2}\left[\frac{r^2}{2\sigma^2} + \frac{1}{2} \log \sigma^2\right] = -\frac{r^2}{2\sigma^4} + \frac{1}{2\sigma^2} = \frac{\sigma^2 - r^2}{2\sigma^4} \tag{22}$$

Solving for when the gradient is zero, gives $\sigma_{opt}^2 = r^2$. However, as our predicted variance is $\sigma_{\boldsymbol{\theta}}^2 = (c_{\boldsymbol{\theta}}^2 + \lambda)(c_{\boldsymbol{\theta}}^2 + \lambda) = c_{\boldsymbol{\theta}}^4 + 2\lambda c_{\boldsymbol{\theta}}^2 + \lambda^2$, it cannot be smaller than $\lambda^2$, and therefore $\sigma_{opt}^2$ is not always obtainable. If $r \geq \lambda$ then $\sigma_{opt}^2$ is obtainable with $c_{\boldsymbol{\theta}}^2 = r - \lambda$. This corresponds to when the loss is one in Figure 2. However, the optimal variance is not obtainable if, $r < \lambda$, as it implies $c_{\boldsymbol{\theta}}^2 < 0$, which is impossible. To better understand what the network does in this case, we look at the gradient of the loss with respect to $c_{\boldsymbol{\theta}}$

$$\frac{\partial}{\partial c_{\boldsymbol{\theta}}}\left[\frac{r^2}{2\sigma_{\boldsymbol{\theta}}^2} + \frac{1}{2} \log \sigma_{\boldsymbol{\theta}}^2\right] = \frac{\partial}{\partial \sigma_{\boldsymbol{\theta}}^2}\left[\frac{r^2}{2\sigma_{\boldsymbol{\theta}}^2} + \frac{1}{2} \log \sigma_{\boldsymbol{\theta}}^2\right]\frac{\partial \sigma_{\boldsymbol{\theta}}^2}{\partial c_{\boldsymbol{\theta}}} = \frac{(\sigma_{\boldsymbol{\theta}}^2 - r^2)(4c_{\boldsymbol{\theta}}^3 + 4\lambda c_{\boldsymbol{\theta}})}{\sigma_{\boldsymbol{\theta}}^4} \tag{23}$$

where the first equality follows from the chain rule. Hence, in the case that $\sigma_{\boldsymbol{\theta}}^2 \geq \lambda^2 > r^2$, the numerator of the gradient can only be zero if $c_{\boldsymbol{\theta}}$ is zero, making $\sigma_{\boldsymbol{\theta}}^2 = \lambda^2$. This corresponds to the (scaled) squared error behavior for small residuals ($r^2 < \lambda^2$) in Figure 2.

### B.5 Target Logits

From Section 2.2, the target categorical for $y = i$ is:

$$S(\boldsymbol{y}) = ((1-t)\boldsymbol{\delta}_i + t\boldsymbol{u})_j = \begin{cases} t/K & i \neq j \\ \frac{(1-t)K+t}{K} & i = j \end{cases} \tag{24}$$

and $S_C^{-1}(\boldsymbol{p}) = \log([p_1, \ldots, p_{K-1}]/p_K)$. Hence, if $i \neq K$, then the observed target logit is:

$$\boldsymbol{y} = S_C^{-1}(S_C(\boldsymbol{y})) = S_C^{-1}((1-t)\boldsymbol{\delta}_i + t\boldsymbol{u})_j = \begin{cases} 0 & i \neq j \\ \mathcal{C} & i = j \end{cases} \tag{25}$$

as $\log\left(\frac{t}{K}/\frac{t}{K}\right) = \log 1 = 0$ and where $\mathcal{C} = \log \frac{(1-t)K+t}{t}$. If $i = K$, then we have for all $j$:

$$\boldsymbol{y} = S_C^{-1}((1-t)\boldsymbol{\delta}_i + t\boldsymbol{u})_j = -\mathcal{C}. \tag{26}$$

### B.6 Gradients for the Heteroscedasitc NN methods

In Heteroscedasitc NN methods, the prediction is the mean of M samples $\bar{\boldsymbol{y}}_c = \frac{1}{M}\sum_{m=1}^{M}\boldsymbol{y}_c^m$, in which we define $\boldsymbol{y}_c^m = e^{\boldsymbol{z}_c + \boldsymbol{\epsilon}^m}/\Sigma_K^m$ and $\Sigma_K^m = \sum_{k=1}^{K}e^{\boldsymbol{z}_k + \boldsymbol{\epsilon}^m}$. Therefore, the gradient of softmax output w.r.t logits is given by:

$$\begin{aligned}
\frac{\partial}{\partial \boldsymbol{z}_j}\frac{1}{M}\sum_{m=1}^{M}\boldsymbol{y}_i^m &= \frac{1}{M}\sum_{m=1}^{M}\frac{\partial \boldsymbol{y}_i^m}{\partial \boldsymbol{z}_j} = \frac{1}{M}\sum_{m=1}^{M}\frac{\partial}{\partial \boldsymbol{z}_j}\frac{e^{\boldsymbol{z}_i + \boldsymbol{\epsilon}^m}}{\Sigma_K^m} \\
&= \frac{1}{M}\sum_{m=1}^{M}\frac{\delta_{ij}e^{\boldsymbol{z}_i + \boldsymbol{\epsilon}^m}\Sigma - e^{\boldsymbol{z}_i + \boldsymbol{\epsilon}^m}e^{\boldsymbol{z}_j + \boldsymbol{\epsilon}^m}}{(\Sigma_K^m)^2} \\
&= \frac{1}{M}\sum_{m=1}^{M}\frac{e^{\boldsymbol{z}_i + \boldsymbol{\epsilon}^m}}{\Sigma_K^m}\left(\frac{\delta_{ij}\Sigma_K^m - e^{\boldsymbol{z}_j + \boldsymbol{\epsilon}^m}}{\Sigma_K^m}\right) \\
&= \frac{1}{M}\sum_{m=1}^{M}\boldsymbol{y}_i^m\left(\delta_{ij} - \boldsymbol{y}_j^m\right) \\
&= \begin{cases} \frac{1}{M}\sum_{m=1}^{M}\boldsymbol{y}_i^m\left(1 - \boldsymbol{y}_j^m\right) & i = j \\ -\frac{1}{M}\sum_{m=1}^{M}\boldsymbol{y}_i^m\boldsymbol{y}_j^m & i \neq j \end{cases}
\end{aligned}$$

Using the last equation, we can compute the derivatives of the log-likelihood with respect to logits as follows:

$$\begin{aligned}
\frac{\partial}{\partial \boldsymbol{z}_j}\sum_{i=1}^{K}\boldsymbol{t}_i\log(\frac{1}{M}\sum_{m=1}^{M}\boldsymbol{y}_i^m) &= \sum_{i=1}^{K}\boldsymbol{t}_i\frac{\frac{\partial}{\partial \boldsymbol{z}_j}\sum_{m=1}^{M}\boldsymbol{y}_i^m}{\sum_{m=1}^{M}\boldsymbol{y}_i^m} \\
&= \sum_{i=1}^{K}\boldsymbol{t}_i\frac{\sum_{m=1}^{M}\boldsymbol{y}_i^m\left(\delta_{ij} - \boldsymbol{y}_j^m\right)}{\sum_{m=1}^{M}\boldsymbol{y}_i^m} \\
&= -\sum_{i \neq j}^{K}\boldsymbol{t}_i\frac{\sum_{m=1}^{M}\boldsymbol{y}_i^m\boldsymbol{y}_j^m}{\sum_{m=1}^{M}\boldsymbol{y}_i^m} \\
&\quad + \boldsymbol{t}_j\left(1 - \frac{\sum_{m=1}^{M}(\boldsymbol{y}_j^m)^2}{\sum_{m=1}^{M}\boldsymbol{y}_j^m}\right) \\
&= \boldsymbol{t}_j - \sum_{i}^{K}\boldsymbol{t}_i\frac{\sum_{m=1}^{M}\boldsymbol{y}_i^m\boldsymbol{y}_j^m}{\sum_{m=1}^{M}\boldsymbol{y}_i^m}
\end{aligned}$$

Assuming a label $\boldsymbol{t} = \boldsymbol{\delta}_c$ is given, we can rewrite the above equation in a simpler form:

$$
\begin{aligned}
\frac{\partial}{\partial \boldsymbol{z}_j} \sum_{i=1}^{K} \boldsymbol{t}_i \log(\frac{1}{M} \sum_{m=1}^{M} \boldsymbol{y}_i^m) =& \boldsymbol{t}_j - \sum_i^K \boldsymbol{t}_i \frac{\sum_{m=1}^{M} \boldsymbol{y}_i^m \boldsymbol{y}_j^m}{\sum_{m=1}^{M} \boldsymbol{y}_i^m} \\
=& \boldsymbol{t}_j - \sum_i^K \boldsymbol{t}_i \frac{\sum_{m=1}^{M} \boldsymbol{y}_i^m \boldsymbol{y}_j^m}{\sum_{m=1}^{M} \boldsymbol{y}_i^m} \\
=& \boldsymbol{t}_j - \frac{\sum_{m=1}^{M} \boldsymbol{y}_c^m \boldsymbol{y}_j^m}{\sum_{m=1}^{M} \boldsymbol{y}_c^m} \\
=& \boldsymbol{t}_j - \sum_{m=1}^{M} \boldsymbol{y}_j^m \frac{\boldsymbol{y}_c^m}{\sum_{m=1}^{M} \boldsymbol{y}_c^m} \\
=& \boldsymbol{t}_j - \frac{1}{M} \sum_{m=1}^{M} \boldsymbol{y}_j^m \frac{\boldsymbol{y}_c^m}{\frac{1}{M} \sum_{m=1}^{M} \boldsymbol{y}_c^m}
\end{aligned}
$$

The derivatives of the log-likelihood, in vector notation, are given by:

$$
\begin{aligned}
\frac{\partial}{\partial \boldsymbol{z}} \left[ \boldsymbol{t}^T \cdot \log(\frac{1}{M} \sum_{m=1}^{M} \boldsymbol{y}^m) \right] =& \boldsymbol{\delta}_c - \frac{1}{M} \sum_{m=1}^{M} \boldsymbol{y}^m \frac{\boldsymbol{y}_c^m}{\frac{1}{M} \sum_{m=1}^{M} \boldsymbol{y}_c^m} \\
=& \boldsymbol{\delta}_c - \frac{1}{M} \sum_{m=1}^{M} S(\boldsymbol{z} + \boldsymbol{\epsilon}^m) \frac{S(\boldsymbol{z} + \boldsymbol{\epsilon}^m)_c}{\frac{1}{M} \sum_{m=1}^{M} S(\boldsymbol{z} + \boldsymbol{\epsilon}^m)_c}
\end{aligned}
$$

## C  Hyperparameters

### C.1  Method-independent Hyperparameters

All methods are implemented in the same code base, using the same network architectures, optimizers, hyperparameter searches, etc. We use a learning rate of 0.0001 for synthetic datasets, 0.001 for MNIST and Clothing1M, and 0.01 for the CIFAR datasets. We show that our method performs well under different optimizers by using gradient descent for the synthetic datasets, Adam for MNIST (batch size 256), and SGD with Nesterov momentum of 0.9 for Clothing1M (batch size 32) and the CIFAR datasets (batch size 128). We use a weight decay of 1e-3 and 5e-4 for Clothing1M and the CIFAR datasets, respectively, but no such regularization for the other datasets. We use an MLP with two hidden layers with 2000 hidden units for the synthetic datasets, a convolutional network (LeNet-5) for MNIST, and residual networks for the CIFAR datasets (WideResNet-28-2) and Clothing1M (ImageNet pre-trained ResNet-50). We train for 2000, 100, 10, and 300 epochs for the synthetic datasets, MNIST, Clothing1M, and CIFAR, respectively. We use 10% of the training set of MNIST and CIFAR as a noisy validation set.

### C.2  Method-dependent Hyperparameter Search

In this section, we go over our thorough hyperparameter search we did for the results in Tables 1 and 2. For each method, we search for method-specific hyperparameters for each noise rate and noise type per dataset. For Het-$\tau$ and Het-$\tau$-$\boldsymbol{\Sigma}_{\mathbf{full}}$, we search for temperatures in $[0.1, 0.5, 1.0, 10.0, 20.0]$, while Het-$\tau$-$\boldsymbol{\Sigma}_{\mathbf{full}}$ also searches over $R$ of the covariance matrix in $[1, 2, 4]$. We choose the range of values to search over is based on the original papers. Our method searches over temperatures and $\lambda$s in $[0.1, 0.5, 1.0]$ for MNIST, but $\lambda$s in $[0.5, 1.0]$ for the CIFAR datasets and Clothing1M. We treat the label smoothing parameter $t$ for LN as fixed, and set it to 0.01 in all experiments. For GCE, we search over $q$ in $[0.1, 0.3, 0.5, 0.7, 0.9]$. For NAN, the search is over $\sigma$ in $[0.1, 0.2, 0.5, 0.75, 1.0]$. For LS, we search for values in $[0.1, 0.3, 0.5, 0.7, 0.9]$. All searches are done with a single seed, and the hyperparameters with the highest noisy validation accuracy at the end

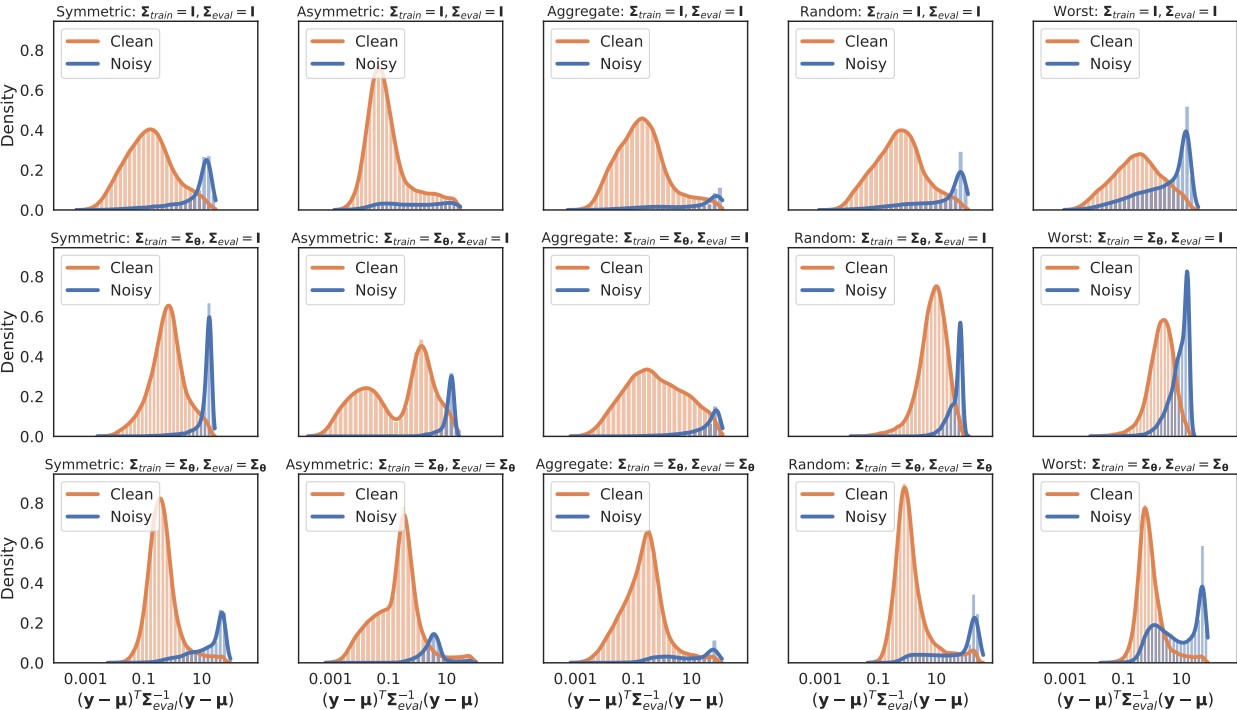

Figure 7: **Loss Attenuation: The Effect of the Covariance Matrix.** For a network trained on MNIST with 40% symmetric noise, we show the distribution of the label-dependent term of the loss with (left) and without (right) the scaling of the covariance matrix for clean and noisy examples separately. The x-axis correspond to the log-scaled loss. We find that the covariance matrix increases the loss of clean examples and reduces the loss of noisy examples.

of training are used to train four more networks with different seeds. The hyperparameters used for different noise rates, noise types, and datasets are shown in Tables 5, and 6.

## D   Natural Datasets

The CIFAR-10N dataset has five new sets of labels for the CIFAR-10 training set, which was generated by having each training image labeled by three different humans. Naturally, this gives rise to three different labeled sets (Random 1-3). The fourth set is generated through majority voting, where ties are broken at random (Aggregate). Finally, the last set of labels (Worst) was created by randomly picking one of the labels that are different from the original training label, if no such a label exists then the original is used. The noisy labels are 18%, 9%, and 40% of all labels for Random, Aggregate and Worst, respectively. CIFAR-100N was created similarly, but with a single human annotator per image, resulting in a noise rate of 40%. Clothing1M is a dataset of one million images of clothes from 14 different classes, automatically labeled based on captions. As there is a large imbalance between the classes, we follow the balancing strategy of Li et al. (2020a). We use the provided validation and test sets. The noise rate is estimated to be 38%.

## E   Implementation Details

We implement our method using the TensorFlow Probability Dillon et al. (2017) library. The Logistic-Normal distribution is implemented as a transformed distribution (the TransformedDistribution class) comprised of a distribution and a transform. For the distribution, we use a multivariate normal distribution (the MultivariateNormalDiagPlusLowRank class). For the transform, we combine a softmax centered and a scale (for temperature) bijector using the Chain bijector class. Conveniently, the loss can then be implemented by

using the built-in method (log_prob) of the transformed distribution class to calculate the logarithm of the pdf.

## F   Connections to Additional Related Work

### F.1   Relationship with Calibration Methods

Both calibration methods and our method are interested in capturing a true probability distribution $p(y|x)$ and in that sense some standard calibration techniques, such as temperature scaling, might be applicable for both. However, there are two key fundamental differences:

- Probabilistic noise models are interested in obtaining the true probability distribution of sampled data (only dependent on data and irrespective of the machine learning method). Calibration methods are by definition tied to a machine learning method and want their probabilistic output to be a true reflection of the "probability of correctness" (only dependent on the machine learning method and irrespective of the data). This can be seen, for instance, by considering that a model that always outputs the uniform distribution is calibrated but minimally accurate.

- Calibration is for unseen (test) data, while noise models are typically for the training data. As such, many calibration methods are often applied as a post hoc approach, tuning calibration metrics on unseen (calibration) data, and then evaluated on unseen (test) data. Furthermore, as the temperature scaling is typically done after training, it has no effect on the training dynamics, while the temperature we use directly affects the observed target locations (see first paragraph in Section 3.2). As noise models, such as ours, are primarily concerned with modelling of the training data, the full distributions (that includes variance due to label noise) can in fact be discarded at test time, see Section 3.3.

### F.2   Relationship with Linear Regression

There is a close relationship between our work and (Multinomial) logistic regression. In fact, what we do with the Logistic-Normal distribution can be seen as a generalization of doing logistic regression as a set of binary regressions, where the generalization is to incorporate label noise. That is,

$$\text{Logistic regression}: S_C(y) = S_C(Wx) \tag{27}$$

$$\text{Our case}: S_C(y) = S_C(Wx + \epsilon) \Leftrightarrow S_C(y) \sim \mathcal{LN}(Wx, \Sigma), \epsilon \sim \mathcal{N}(0, \Sigma) \tag{28}$$

where W are the parameters of the single linear layer. In the logistic regression case, we have no noise, and the resulting likelihood is a categorical distribution. However, in our case, due to the noise being normally distributed, the resulting likelihood is a Logistic-Normal distribution, which is a distribution over categorical distributions. This makes it possible for us to explain differences between potentially noisy labels $\tilde{p}(y|x)$, with our approximation of the true categorical distribution $p_{\boldsymbol{\theta}}(y|x)$ as label noise, akin to the regression case.

### F.3   Relationship with Linear Discriminant Analysis

We see standard LDA as a fundamentally different method as it puts an explicit distribution on the samples belonging to each class, which we do not.

To simplify the analysis, let's consider a fixed mapping from the input space $x$ to logit space, i.e., LDA operates on $h(x)$ instead of $x$, and LN has $\mu(x) = h(x)$, and let's call these the "features".

LDA assumes that the features of all examples of a particular class are samples from the same normal distribution. The variance of the features of examples of the same class, could be interpreted as per-class label noise. Hence, we expect LDA to not be robust to input/feature/heteroscedastic noise, as a single example with a noisy label, could dramatically affect the learnt location for the distribution for that class.

In LN, we have fixed positions for where the class clusters should be, i.e., the target logit locations. LN assumes each feature corresponds to the true logit position for that example, and the difference between

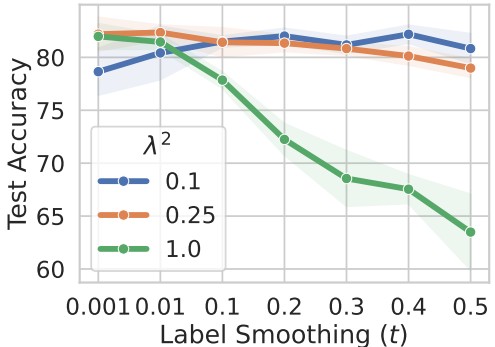

Figure 8: **Effect of Label Smoothing on Robustness of LN.** The mean and standard deviation of test accuracy for various label smoothing parameters $t$ on CIFAR-10 with 40% symmetric noise. For small $\lambda^2$, label smoothing has little effect on robustness. Interestingly, for $\lambda^2 = 1$ the networks overfit to the noisy labels more for increasing $t$ values instead of improving robustness.

the feature and observed target location is due to normally distributed label noise. Hence, we treat each observed target as a sample from a normal distribution centered on its corresponding feature. If we think of the features of all examples of a particular class, as we did for LDA, then some might form a cluster around the given target logits, while some features might be closer to other target logit locations. Consider such a feature that is far away from the observed (noisy) target logit location. Our method would then have a large variance to make the target logit more likely, while still keeping the correct feature location the same.

To summarize, LDA estimates per-class normal distributions such that the features of the examples of the corresponding class distribution are likely samples. In contrast, LN estimates per-example normal distributions (up to constants) centered on each feature, such that the corresponding target logit is a likely sample. We expect LDA to not be robust against noisy labels, as a single example with a noisy label could completely shift the location of the per-class distribution for that particular noisy label. Our model is affected less, as noisy examples could still have the correct location, while having a larger variance instead. Furthermore, at test time, our method predicts per-example distributions, while LDA evaluates the per-class distributions for each input.

## G   Additional Experiments

If not otherwise stated, all the results in the tables in this section reports the mean and standard deviation for the test accuracy over five runs with different seeds.

### G.1   On the Robustness of the Logistic-Normal Likelihood Due to Label Smoothing

As the Logistic-Normal distribution is not defined on the border of the probability simplex, we proposed to use label smoothing to solve this issue. Interestingly, Lukasik et al. (2020) showed that label smoothing itself helps with robustness against label noise. Therefore, a natural question is how important label smoothing is for the robustness of our method. In Figure 8, we show the mean and standard deviation of test accuracy for various settings of the label smoothing parameter $t$ (defined in Section 2.2) for our method trained on 40% symmetric noise on CIFAR-10. We find that for $\lambda^2$ of 0.1 and 0.25, the label smoothing parameters have little effect on the test accuracy, suggesting that label smoothing is not having a big effect on the robustness of our method. Interestingly, for $\lambda^2 = 1$, increasing $t$ degrades the test accuracy instead of improving it. We believe this is similar to the temperature, that a too high value for $t$ makes most of the residuals be below the loss attenuation threshold and therefore the loss behaves like a standard mean squared error loss and overfits.

Table 4: **Calibration on the CIFAR-10N Training Set.** CIFAR-10N provides three labels per training example, which we see as three samples from the true $p(y|x)$. Instead of measuring calibration with a single (potentially noisy label), we propose to measure calibration on the training set by taking the average negative log-likelihood (NLL) of each of the provided labels. We observe that LN has significantly lower average NLL and is therefore better calibrated.

| Method | NLL |
|---|---|
| Het-$\tau$-$\boldsymbol{\Sigma_{\text{full}}}$ | $1.66 \pm 0.03$ |
| CE | $1.60 \pm 0.01$ |
| GCE | $1.58 \pm 0.04$ |
| NAN | $1.54 \pm 0.05$ |
| LN | $\mathbf{1.18 \pm 0.01}$ |

## G.2 Training Set Calibration

Calibration metrics are typically used as scalar surrogates to measure the difference between the true $p(y|x)$ and the prediction $p_{\boldsymbol{\theta}}(y|x)$ on unseen data. This is typically done using, $\tilde{p}(y|x)$ as we don't usually have access to $p(y|x)$. However, interestingly, as CIFAR-N provides three labels from human annotators per image in the CIFAR-10 dataset, we can see these as three samples from $p(y|x)$ and have a better estimate of the true label via:

$$p(y|x) = \lim_{M \to \infty} \frac{1}{M} \sum_{m=1}^{M} \text{onehot}(y^{(m)}) \approx \frac{1}{3}(\text{onehot}(y^{(1)}) + \text{onehot}(y^{(2)}) + \text{onehot}(y^{(3)})) = \tilde{p}(y|x) \quad (29)$$

where $y^{(m)} \sim p(y|x)$. Hence, we propose to measure the training set calibration of models trained on the CIFAR-10 dataset (no added noise), by evaluating the standard negative log-likelihood of the predicted categorical distribution using the three labels provided by CIFAR-N. More specifically, the per-example NLL is calculated as $\frac{1}{3}(-\log p_{\boldsymbol{\theta}}(y = y^{(1)}|x) - \log p_{\boldsymbol{\theta}}(y = y^{(2)}|x) - \log p_{\boldsymbol{\theta}}(y = y^{(3)}|x))$, and all per-example NLLs of the training set are then averaged. See the table below for the results, which is the mean and standard deviation of the five networks of models trained with no synthetically added label noise in Table 1.

As with the standard NLL calibration metrics, lower values indicate the model has put more confidence into these classes, which is therefore desired. From the results in Table 4, we find that our method has significantly lower NLL on the training set than the other methods.

## G.3 Residual Histograms

In this section, we show similar histograms as in Figure 6, but for more noise types: symmetric noise and aggregate, random 2, and worst from CIFAR-10N. The setup is the same, to train two networks by minimizing the negative log-likelihood of Logistic-Normal likelihoods with different covariance matrices ($\boldsymbol{\Sigma}_{train}$) and evaluate the label-dependent term of Equation 5, with either $\boldsymbol{\Sigma}_{eval}$ equal to $\boldsymbol{I}$ or $\boldsymbol{\Sigma_{\theta}}$ at the end of training. See Figure 7. Comparing $\boldsymbol{\Sigma}_{train} = \boldsymbol{\Sigma}_{eval} = \boldsymbol{I}$ (top row) with $\boldsymbol{\Sigma}_{train} = \boldsymbol{\Sigma_{\theta}}$, $\boldsymbol{\Sigma}_{eval} = \boldsymbol{I}$ (middle row) that the former reduces the residuals of more of the noisy examples than the latter, which indices learning $\boldsymbol{\Sigma}$ is more robust. Furthermore, comparing $\boldsymbol{\Sigma}_{train} = \boldsymbol{\Sigma_{\theta}}$, $\boldsymbol{\Sigma}_{eval} = \boldsymbol{I}$ (middle row) with $\boldsymbol{\Sigma}_{train} = \boldsymbol{\Sigma_{\theta}}$, $\boldsymbol{\Sigma}_{eval} = \boldsymbol{\Sigma_{\theta}}$ (bottom row), we find that $\boldsymbol{\Sigma_{\theta}}$ increases the loss for some clean examples and reduces the loss of high-residual examples, both clean and noisy ones.

Table 5: **Hyperparameters used for Synthetic Noise on MNIST and CIFAR datasets.** A hyperparameter search over method-specific hyperparameters is done, and the best values are shown here. For Het-$\tau$, we report the temperature, for Het-$\tau$-$\mathbf{\Sigma_{full}}$ the temperature and the number of factors ($R$), for GCE $q$, for NAN $\sigma$, for LS $t$, and for LN the temperature ($\tau$) and $\lambda$.

| Dataset | Method | No Noise | Symmetric Noise Rate | | | Asymmetric Noise Rate | | |
|---|---|---|---|---|---|---|---|---|
| | | 0% | 20% | 40% | 60% | 20% | 30% | 40% |
| MNIST | Het-$\tau$ | 0.5 | 0.1 | 20 | 10 | 0.5 | 0.5 | 0.1 |
| | Het-$\tau$-$\mathbf{\Sigma_{full}}$ | [1.0, 1] | [0.1, 4] | [0.1, 4] | [10, 4] | [10, 2] | [0.5, 1] | [10, 1] |
| | NAN | 0.2 | 1.0 | 1.0 | 1.0 | 1.0 | 0.75 | 1.0 |
| | GCE | 0.3 | 0.7 | 0.9 | 0.9 | 0.7 | 0.5 | 0.3 |
| | LS | 0.5 | 0.3 | 0.9 | 0.9 | 0.1 | 0.5 | 0.7 |
| | LN | [1.0, 1.0] | [1.0, 1.0] | [1.0, 0.5] | [0.5, 0.1] | [0.5, 0.1] | [0.5, 0.1] | [0.5, 0.1] |
| CIFAR-10 | Het-$\tau$ | 0.5 | 10 | 20 | 10 | 10 | 10 | 20 |
| | Het-$\tau$-$\mathbf{\Sigma_{full}}$ | [0.5, 4] | [20, 4] | [20, 1] | [10, 2] | [20, 2] | [0.1, 2] | [0.5, 2] |
| | NAN | 0.2 | 0.5 | 0.75 | 0.75 | 0.5 | 0.1 | 0.2 |
| | GCE | 0.1 | 0.9 | 0.9 | 0.9 | 0.9 | 0.9 | 0.1 |
| | LS | 0.7 | 0.9 | 0.9 | 0.9 | 0.5 | 0.3 | 0.9 |
| | LN | [0.1, 0.5] | [0.5, 0.5] | [1.0, 0.5] | [1.0, 0.5] | [0.5, 0.5] | [0.5, 0.5] | [0.5, 0.5] |
| CIFAR-100 | Het-$\tau$ | 20 | 20 | 10 | 10 | 20 | 20 | 10 |
| | Het-$\tau$-$\mathbf{\Sigma_{full}}$ | [0.5, 4] | [10, 4] | [10, 4] | [10, 1] | [20, 2] | [20, 1] | [20, 1] |
| | NAN | 0.1 | 0.2 | 0.2 | 0.2 | 0.1 | 0.2 | 0.1 |
| | GCE | 0.5 | 0.5 | 0.5 | 0.5 | 0.7 | 0.7 | 0.5 |
| | LS | 0.1 | 0.9 | 0.7 | 0.7 | 0.9 | 0.7 | 0.9 |
| | LN | [0.1, 0.5] | [1.0, 0.5] | [1.0, 0.5] | [1.0, 0.5] | [0.5, 0.5] | [0.5, 1.0] | [0.5, 1.0] |

Table 6: **Hyperparameters used for Natural Noise** A hyperparameter search over method-specific hyperparameters is done, and the best values are shown here. For Het-$\tau$, we report the temperature, for Het-$\tau$-$\mathbf{\Sigma_{full}}$ the temperature and the number of factors ($R$), for GCE $q$, for NAN $\sigma$, for LS $t$, and for LN the temperature ($\tau$) and $\lambda$.

| Method | CIFAR-10N | | | | | CIFAR-100N | Clothing1M |
|---|---|---|---|---|---|---|---|
| | Random 1 | Random 2 | Random 3 | Aggregate | Worst | | |
| Het-$\tau$ | 10 | 20 | 20 | 0.5 | 20 | 20 | 0.1 |
| Het-$\tau$-$\mathbf{\Sigma_{full}}$ | [10, 4] | [0.1, 1] | [20, 1] | [20, 1] | [10, 4] | [10, 1] | [0.1, 1] |
| NAN | 0.75 | 0.5 | 0.5 | 0.5 | 0.75 | 0.2 | 0.2 |
| GCE | 0.9 | 0.7 | 0.9 | 0.5 | 0.9 | 0.5 | 0.9 |
| LS | 0.5 | 0.9 | 0.5 | 0.9 | 0.7 | 0.9 | 0.5 |
| LN | [0.5, 0.5] | [1.0, 0.5] | [0.5, 0.5] | [1.0, 1.0] | [0.5, 0.5] | [0.5, 0.5] | [1.0, 1.0] |

