# OpenReview forum: "Logistic-Normal Likelihoods for Heteroscedastic Label Noise"
_TMLR — Accepted by TMLR_

### Review · Reviewer_vzPd · 2023-05-28

**Summary Of Contributions:**

Drawing inspiration from how regression methods naturally account for heteroscedastic noise by treating the problem as a maximum likelihood estimation with Gaussian noise, this paper introduces a classification counterpart: estimation with logistic-normal likelihoods. The proposed method exhibits an interpretation of loss attenuation, grounded in the covariance matrix, similar to the regression case where each sample's loss is weighted based on the variance of the considered normal distribution. The paper acknowledges and addresses some unique challenges, such as the inability to consider $y_i$ that is mapped to a one-hot vector, and the handling of classes due to the softmax centering issue. The experimental findings showcase the competitiveness of the proposed method, not only for conventional classification problems but also for those involving noisy-labels.

**Audience:**

Yes

**Broader Impact Concerns:**

I do not have strong ethical concerns of this paper.

**Claims And Evidence:**

Yes

**Requested Changes:**


- Add explanation of the hyper-parameter $t$ used in experiments.
- Add definition of $S$ if it is different from $S_C$.
- Add explanation of the bi-tempered baseline.

Additionally, in my opinion, adding experiments and discussions about label smoothing will be interesting. I wrote my detailed comment in the "Strengths and Weaknesses" section.


**Strengths And Weaknesses:**

Strengths:
- The paper shows an interesting extension of loss attenuation in regression to classification.
- The paper studies the loss attenuation from investigating the gradients.
- The paper provides experimental evidence about how the proposed method is more robust to label noise compared with baselines for synthetic (class-dependent) noise and natural noise.
- The paper provides code used in the experiments (note: I did not read the code carefully as part of my review.)

Weaknesses/questions:
- I found it challenging to locate the values of the label smoothing hyperparameter $t$ utilized in the experiments. It would be helpful if the authors can explain where it is shown.
- Given its vital role in the proposed method, perhaps an ablation study of this hyperparameter could provide more clarity and will help readers understand the method better.
- The usual cross-entropy loss minimization has a similar issue of this, where we try to fit the output of the softmax to a one-hot classification label. One way to avoid this is to use label smoothing. The technique referenced in Section 3.2 appears to align with this approach. With reports indicating label smoothing's usefulness in ordinary classification problems and label-noise problems (e.g., Lukasik et al., ICML 2020), it's not immediately evident whether the loss attenuation based on the proposed method enhanced generalization, or if the label smoothing component did, or if both contributed. Thus, a comparison with another baseline (label smoothing + cross entropy) could be helpful.
- The bi-tempered method is used in experimental results shown in Tabel 5, 6, and 7. However, this is not explained in the paper. While it's possible that I overlooked it, I did not find it explicitly mentioned among the baselines in Section 5.2. Perhaps this could be clarified?
- The notation $S$ occurs in Eq.13, Eq.14, Eq.24, and in subsequent places. I'm curious if $S$ is equivalent to $S_C$ as defined in Section 2.2? Could you provide some clarification on this?

---

> ### Author Response · Authors · 2023-06-01
>
> Hi Reviewer vzPd.
>
> We greatly appreciate your time and effort in making such a constructive and thorough review. We are happy you found our method interesting and that our gradient analysis, experiments, and code were deemed strengths of our work. We believe the reviewer has many good questions, which we answer below.
>
> **Label smoothing.** In Section 3.2, we mention that the label smoothing hyperparameter $t$ is set to a fixed small value, but indeed, we never specify the value. Thanks for noticing this. In all our experiments, we have $t=0.01$. We are currently running an ablation study to better understand the impact of this hyperparameter. Furthermore, as our method uses label smoothing, the work you mention (CE+label smoothing) is indeed a relevant baseline, which we will add.
>
> **Old baseline in the appendix.** An early reader of our work proposed comparing LN with a robust loss function method called Bi-tempered [1]. However, in the end, we decided to not include it as a baseline as the GCE method (which is also a method from the robust loss functions) performs significantly better or as good as the Bi-tempered method. Unfortunately, we missed removing mentions of this method from some tables in the appendix. Thanks for noticing this, and this is now fixed.
>
> **Is $S$ equivalent to $S_C$?** No, there is a small but important difference related to the last component. S is the standard softmax (see paragraph under Equation 6), i.e., $S(\boldsymbol{z}) = \text{softmax}([z_1, \dots, z_{K-1}, z_{K}])=[e^{z_1}, \dots, e^{z_K}]/\sum_{i=1}^{K}e^{z_i}$, while the softmax centered (defined in Section 2.2), is $S_C(\boldsymbol{z}) = \text{softmax}([z_1, \dots, z_{K-1}, 0])$. Note that $S_C$ is a bijection, but $S$ is not as $S(\boldsymbol{z})=S(\boldsymbol{z}+\boldsymbol{c})$, where $\boldsymbol{c}$ is any K-dimensional vector with the same value in each component. The bijectivity of $S_C$ makes it possible to use the change of variables technique for probability densities to derive the corresponding density (LN) of the transformed Gaussian random variable in Equation 3, see Appendix B.1. We believe the difference between $S$ and $S_C$ could be made clearer in the paper. Therefore, we plan to modify the paragraph in Section 2.2 to first define the standard softmax function and then the softmax centered. Additionally, we will add that $S$ is the standard softmax function under Equation 13. We believe these modifications should make the difference between $S$ and $S_C$ clearer. Please let us know if you have a better suggestion.
>
> We are currently running experiments and updating our paper based on your feedback. We will upload the revised version on OpenReview once all three reviews are in; as per TMLR suggestion.
>
> [1] Amid, E., Warmuth, M. K., Anil, R., & Koren, T. (2019). Robust bi-tempered logistic loss based on bregman divergences. Advances in Neural Information Processing Systems, 32.

---

### Review · Reviewer_6hsn · 2023-06-06

**Summary Of Contributions:**

The authors propose performing regression on the logits in classification with a neural network, with heteroskedastic noise that is also the output of a neural network.
This is done by adding uniform noise to the labels and pulling-back the (noisy) class labels via the (centered) soft-plus. The authors compare this approach to prior approaches involving
heteroskedastic noise for classification on several real and synthetic tasks, with most of the real data tasks being image tasks.

**Audience:**

Yes

**Broader Impact Concerns:**

I don’t think a broader impact statement is needed for the paper.

**Claims And Evidence:**

Yes

**Requested Changes:**

- The simplex ($\Delta^K$) should be defined, with some care around the boundary. In particular, it is not true that the map $S_C$ is bijective between the closed simplex and $R^{K-1}$, but from the interior of the simplex to $R^{K-1}$. This is an important detail since, as you later note $\tilde{p}(y|x)$ is on the boundary of the simplex. Discussion of this point should come earlier. Relatedly, the density in equation 4 isn’t well-defined on the boundary of the simplex, because of the term with $\tilde{p}(y|x)$ in the denominator. I think these issues should be fixed for the paper to be accepted.
- The legend in figure 4 is too small.


**Strengths And Weaknesses:**

### Strengths:
- The method is reasonably simple to understand and implement.
- Writing is generally good and the description of the method can be followed.

### Weaknesses:
- Several claims are made that are slightly wrong or at least not well-defined.
- The method introduces several hyper parameters $t, \tau, \lambda$ that must be tuned.
- It isn’t clear on what sorts of real datasets I would prefer using this method to GCE. In practice, it seems that GCE often works as well or better, and there isn't a concrete description of types of real problems where this method should be preferred.

---

### Review · Reviewer_Nxf4 · 2023-06-10

**Summary Of Contributions:**

This work studied the logit-normal distribution for modeling the probability of noisy labels.
The proposed method is based on a neural network prediction of logit/covariance and maximum likelihood.
The method was evaluated on synthetic data and some image classification datasets.

**Audience:**

Yes

**Broader Impact Concerns:**

The author did not discuss broader impacts in the paper.


**Claims And Evidence:**

No

**Requested Changes:**

- The author said the method is a maximum a posteriori estimation, but in fact, the proposed method is simply maximum likelihood. No prior was given. Even if we can say ML is MAP given an uninformative prior, their focuses are still different.
- "As yi is a single sample from p(y|xi), we see it as a noisy version, p̃(y|xi) = δyi , of the true label distribution p(y|xi)" This sentence is not understandable for me.
- ϵp(x) = p̃(y|x) - p(y|x): No, the substraction is not defined. A simplex is included in a vector space, but not all vector operations are defined in a simplex. The difference of probabilities is meaningless in this context.
- "our loss is almost identical to a multivariate version of the regression loss": There is no _the_ regression loss. Besides the Gaussian model, we can also use, e.g., Laplace or Gumbel. This is also why the choice of distribution should be justified for a certain application.
- Why does the proposed method use gradient-based optimization for the covariance matrix but not the optimal one directly (Eq. (7))?

**Strengths And Weaknesses:**

## Strengths

- The approach of using a distribution over the simplex is reasonable.
- The proposed method is simple and easy to implement.

## Weaknesses

- The choice of logit-normal distribution was not well justified. Even Atchison and Shen said in their paper that "Most statisticians ... would start and end with ... Dirichlet distributions." Why not Dirichlet? The author did not explain why it is reasonable to assume that the logits are normally distributed nor verify whether real data follows the logit-normal distribution.
- The estimation of covariate matrices could be unreliable. Since there is usually one label for each input, it is unlikely that the "per-sample heteroscedastic noise covariate matrices" can be accurately estimated from the data. Besides, if one uses neural networks to predict them, the outputs might be too smooth. The author only used the test accuracy as a metric but did not directly verify if the covariate matrices can be correctly estimated (not even on synthetic data).
- The author provided the code, which is nice, but the code is not well documented and quite unreadable. I cannot pin down the location the proposed method is defined, so I cannot verify the implementation.

---

> ### Author Response · Authors · 2023-06-14
>
> Hi Reviewer Nxf4.
>
> Thank you for taking the time to review our work. We are glad that you found our approach of working with distributions over the simplex reasonable and our method simple and easy to implement. Next, we address your comments and questions.
>
> **Justification for using the Logistic-Normal distribution.**
>
>
> Indeed, when Atchison and Shen discussed the Logistic-Normal distribution in their 1980 paper, they recognized that most statisticians at the time only _knew_ of the Dirichlet distribution
>
> “Most statisticians if asked to list the distributions they know over the unit interval (0, 1) and its generalization, the d-dimensional simplex $S^d$, would start and end with the class of beta distributions and their higher-dimensional counterpart, the Dirichlet distributions.”
>
> Why are we not using the Dirichlet distribution? Our goal was to achieve the same loss attenuation properties of the Gaussian likelihood in regression for classification. We believe this justifies the use of Gaussian distributions in logit space, which with the combination of the softmax centered corresponds to the Logistic-Normal distribution. We deem it unlikely that the same loss attenuation behavior can be obtained by using a Dirichlet distribution, as it is not flexible enough (in terms of parameters) to do this. In Section 3 of Atchison and Shen [1], they discuss the possibilities to approximate Dirichlet distributions with Logistic Normal distributions, as well as comparing properties between the two distribution types. In fact, we planned to have the negative log-likelihood of the Dirichlet as a baseline, but we could not make the training stable.
>
> “The author did not explain why it is reasonable to assume that the logits are normally distributed nor verify whether real data follows the logit-normal distribution.”
>
> The statistician George Box famously said: “All models are wrong, but some are useful.” We believe this applies here, as assuming normally distributed logits have been shown to be useful in a range of works (e.g., [2,3,4]). Furthermore, we believe our own work has shown the usefulness of the Logistic-Normal distribution for robustness to label noise.
>
> **Reliable estimation with a single label sample.**
>
> Indeed, reliable estimation of the covariance matrix of a random variable requires several samples. However, before getting there, we will first argue that the same holds for the mean/mode, which is our goal to reliably estimate.
>
> In classification, it is standard practice to use a single label sample to estimate the _mode_ (most likely class) of $p(y|x)$ with a softmax neural network using the negative log-likelihood as loss. Reliable estimation of the mode of $p(y|x)$ with a single sample requires having $p(y|x) = \delta_{j(x)} = \tilde{p}(y|x)$ for some class ID $j(x)$, i.e., the underlying generation process of the labels is noise-free.
>
> In practice, there is aleatoric uncertainty in $p(y|x)$ due to annotation mistakes, and similarities between classes, etc., which violates this assumption. This is problematic, as if $p(y|x)$ is not $\delta_{j(x)}$, but its mode is the $j$th class, then the single label class ID sample $q(x) \sim p(y|x)$ could be different from $j(x)$. Therefore, the standard classification procedure would lead to the wrong, unreliable estimate of the mode, i.e., overfit to the noisy label. This necessitates a different approach.
>
> In our work, we try to improve the estimation of the mode of $p(y|x)$ from a single label sample by removing the incorrect assumption that $p(y|x)$ is noise-free. Instead, we take the opposite approach and assume $p(y|x)$ is never completely noise-free, i.e., $p(y|x) \in \mathring{\Delta}^{K-1}$ (simplex excluding borders). Note that, by choosing to use the softmax function, which cannot map to the borders of the simplex, this assumption is already implicitly made. With this assumption, we make use of a bijection (softmax centered) to map labels from $\mathring{\Delta}^{K-1}$ to an unconstrained regression space $\mathbb{R}^{K-1}$, where we can take advantage of classical regression techniques for modelling label noise.
>
> To get the desirable loss attenuation properties from the Gaussian likelihood in regression, we model the given noisy label logit $S^{-1}_C(\tilde{p}(y|x))$ as the true label logit  $S^{-1}_C(p(y|x)$) with additive Gaussian noise. As both the true label logit and the covariance matrix are unknown, we estimate them with a neural network, which amortizes the difficulty of estimating mean and variance. But note that our goal is not to accurately model the covariance matrix, but rather to improve the reliability of our estimate of the mode.

---

> > ### Author Response · Authors · 2023-06-14
> >
> > **Reliable estimation with a single label sample. [cont]**
> >
> > With the standard approach to classification, the only way to increase the likelihood is to increase the probability of the given mode specified by the given class ID label. With the Logistic-Normal likelihood, we have an alternative way to increase the likelihood by increasing the variance in the direction of the noisy label, and therefore reducing the need to change the mode. Based on the gradient analysis, this is exactly what happens when learning with the Gaussian likelihood in regression, and the Logistic-Normal likelihood in classification, i.e., the loss attenuation property.
> >
> > We hope this clarifies that it is not necessary to reliably estimate the true covariance in the likelihood to improve the robustness of the estimation of the mode.
> >
> > **Why not use the optimal Sigma in Equation 7 directly?**
> >
> > The optimal sigma maximises the likelihood of the observed label $\tilde{p}(y|x)$ for a given $\mu$. If one directly uses $\Sigma_{opt}$ when the network has not learnt good enough estimates of $\mu$, then this leads to slow convergence as all examples are being loss attenuated directly instead of, ideally, only the noisy examples.
> >
> > In some early experiments, we achieved good results by transitioning $\Sigma$ from the identity matrix $I$ to $\Sigma_{opt}$ over the course of training. However, some examples get better predictions for $\mu$ earlier than others during training, and also this requires one to design a transitioning schedule that likely depends on some hyperparameter. We think learning $\Sigma$ to be a better solution.
> >
> >
> >
> > **The quality of estimating covariance matrices with neural networks**
> >
> > It is true that using neural networks to predict the covariance matrices could lead to problems, e.g., too smooth. We argue in the paper that this could be a reason why LN does not work as well for the uniform noise case, which has less structure to it than the asymmetric noise.
> >
> > We do use the test accuracy to measure robustness in most of our experiments. However, in Figure 6, we analyze the effect of the estimated covariance matrices in terms of residuals, showing that it increases and decreases the contributions of the low- and high-residual examples, respectively. This is the desired loss attenuation behavior.
> >
> >
> >
> > **Documentation and readability of code.**
> >
> > Our code follows the same structure as all other implementations in the Uncertainty Baselines library [5] provided by Google. We will improve the documentation for the official release of the code.
> > The most relevant sections of the code are:
> > * _baselines/cifar/logitnormal-factors-11cls-rank1.py_: A function to create the Logistic-Normal distribution is on line 119, and used in the loss on line 422.
> > * _baselines/cifar/wide_resnet_logitnormal_predLabel.py_: We modify the wide ResNet architecture on line 257 by adding the scale/covariance head.
> >
> >
> >
> > **Clarification regarding the interpretation of $\tilde{p}(y|x)$ as a “noisy version” of $p(y|x)$**
> >
> >
> > Thanks for pointing out that this was unclear. One can see the true label distribution as $p(y|x) = \mathbb{E}_{q \sim p(y|x)}[\delta_q]$. We can therefore see $\tilde{p}(y|x) = \delta_q, q \sim p(y|x)$, as a single sample Monte Carlo estimate of the expectation.
> > We believe this would be clearer if we replaced
> >
> > “As $y_i$ is a single sample from $p(y|x_i)$, we see it as a noisy version, $\tilde{p}(y|x_i)=\delta_{y_i}$, of the true label distribution $p(y|x_i)$.”
> >
> > with
> >
> > “As $y_i$ is a single sample from $p(y|x_i)$, we see $\tilde{p}(y|x_i)=\delta_{y_i}$ as a crude estimate of the true label distribution $p(y|x_i)$.”
> >
> >
> >
> >
> > **Clarification regarding ML or MAP estimation.**
> >
> > If we only used the negative log-likelihood of the data given the parameters $\theta$ (Equation 5), we would be doing maximum likelihood estimation of $\theta$. However, as we also incorporate the log prior over $\theta$ in the loss (mentioned just above the same equation), we are doing maximum a posteriori estimation.
> >
> >
> >
> > **Clarification regarding label noise definition.**
> >
> > Thanks for pointing out that subtraction is ill-defined in the probability simplex. We should have been more careful in our wording. The correct definition of the noise we are modelling is $\epsilon(x) = S_C^{-1}(\tilde{p}(y|x)) - S_C^{-1}(p(y|x))$. However, in Section 2.1, we have not defined the softmax centered function yet. We propose to rephrase this to
> >
> > “Occasionally, sampling $p(y|x)$ gives an unlikely sample $y$, e.g., when probable errors are made in the annotation, causing a large difference between $p(y|x)$ and $\tilde{p}(y|x)$. We aim to model this label noise (difference between $p(y|x)$ and $\tilde{p}(y|x)$) in the pre-softmax logit space.”

---

> > > ### Author Response · Authors · 2023-06-14
> > >
> > > **Clarification regarding “the” regression loss.**
> > >
> > > The full quote is the following
> > >
> > > “Our goal was to find a classification loss with similar loss attenuation to the loss for regression. Comparing Equations 2 and 5, we see that our loss is almost identical to a multivariate version of the regression loss.”
> > >
> > > To reduce the risk of misunderstanding “the regression loss” as stating there is only one possible regression loss (which is indeed incorrect), we propose rephrasing the second sentence to
> > >
> > > “We see that our loss in Equation 5 is almost identical to a multivariate version of the regression loss in Equation 2.”
> > >
> > >
> > > **Final Remark**
> > >
> > > We noticed that the reviewer marked the claims and evidence criteria as unsatisfactory. We believe we have addressed all raised concerns in our response, including the ones regarding the provided evidence. If in the reviewer’s view any such concern remains in spite of our response, please precisely specify which claims lack evidence, so we can directly address it.
> > >
> > >
> > > **References**
> > >
> > > [1] Atchison, J., & Shen, S. M. (1980). Logistic-normal distributions: Some properties and uses. Biometrika, 67(2), 261-272.
> > >
> > > [2] Kendall, A., & Gal, Y. (2017). What uncertainties do we need in bayesian deep learning for computer vision?. Advances in neural information processing systems, 30.
> > >
> > > [3] Liu, J., Lin, Z., Padhy, S., Tran, D., Bedrax Weiss, T., & Lakshminarayanan, B. (2020). Simple and principled uncertainty estimation with deterministic deep learning via distance awareness. Advances in Neural Information Processing Systems, 33, 7498-7512.
> > >
> > > [4] Williams, C. K., & Rasmussen, C. E. (2006). Gaussian processes for machine learning (Vol. 2, No. 3, p. 4). Cambridge, MA: MIT press.
> > >
> > > [5] Nado, Z., Band, N., Collier, M., Djolonga, J., Dusenberry, M. W., Farquhar, S., ... & Tran, D. (2021). Uncertainty Baselines: Benchmarks for uncertainty & robustness in deep learning. arXiv preprint arXiv:2106.04015.

---

### Author Response · Authors · 2023-06-14
**The revised version of our paper is now available.**

Dear all,

We would like to thank the reviewers for their valuable feedback so far, which has already improved our work. The changes made in this revision are as follows:

**vzPd**
-  Specified the value of the fixed label smoothing parameter $t$ in a footnote in Section 2.2.
-  Added a sensitivity analysis of $t$ in Appendix G.1.
-  Added label smoothing as a baseline for all datasets, see Tables 1 and 2.
-  Removed old baseline from the Appendix.
-  Updated the softmax centered paragraph in Section 2.2 to make the difference between $S$ and $S_C$ clearer (and clarify $S$ under Equation 13).

**6hsn**
- Clarified the co-domain of softmax centered and its inverse in Section 2.2, and updated the paper to use the new notation for the interior of the simplex.
- Clarified that the Logistic-Normal distribution is defined in the interior of the simplex in the Likelihood function paragraph in Section 2.2.
- Moved discussion of smoothed targets from Section 3.2 to Section 2.2.
- Updated Figure 4 to have larger legend.

**Nxf4**
- Clarified our view of $\tilde{p}(y|x)$ in Section 2.2.
- Rephrased our definition of label noise to not depend on subtractions in simplex in Section 2.2.
- Clarified the sentence in Section 2.3 comparing the regression loss in Equations 2 with the loss in Equation 5.

---

### Decision · Action_Editors · 2023-07-15

**Recommendation:** Accept with minor revision

**Comment:**

The authors propose performing regression on the logits in classification with a neural network, with heteroskedastic noise that is also the output of a neural network. This is done by adding uniform noise to the labels and pulling-back the (noisy) class labels via the (centered) soft-plus. The authors compare this approach to prior approaches involving heteroskedastic noise for classification on several real and synthetic tasks, with most of the real data tasks being image tasks.

The method is reasonably simple to understand and implement. Writing is generally good and the description of the method can be followed. However, the authors are missing some important references in Memorization Effects category (e.g., MentorNet and Co-teaching series). Therefore, based on three qualified reviews, this paper can be accepted with minor revision and the authors are encouraged to merge the comments into their update versions.

**Audience:**

Yes

**Claims And Evidence:**

Yes